# Stretchable phosphorescent polymers by multiphase engineering

Nan Gan[1], Xin Zou[1], Zhao Qian[2], Anqi Lv[3], Lan Wang[3], Huili Ma[3], Hu-Jun Qian[2], Long Gu [1,4] ✉, Zhongfu An [3] ✉ & Wei Huang [1,3] ✉

Stretchable phosphorescence materials potentially enable applications in diverse advanced fields in wearable electronics. However, achieving room-temperature phosphorescence materials simultaneously featuring long-lived emission and good stretchability is challenging because it is hard to balance the rigidity and flexibility in the same polymer. Here we present a multiphase engineering for obtaining stretchable phosphorescent materials by combining stiffness and softness simultaneously in well-designed block copolymers. Due to the microphase separation, copolymers demonstrate an intrinsic stretchability of 712%, maintaining an ultralong phosphorescence lifetime of up to 981.11 ms. This multiphase engineering is generally applicable to a series of binary and ternary initiator systems with color-tunable phosphorescence in the visible range. Moreover, these copolymers enable multi-level volumetric data encryption and stretchable afterglow display. This work provides a fundamental understanding of the nanostructures and material properties for designing stretchable materials and extends the potential of phosphorescence polymers.

Stretchable and flexible materials are highly desirable for developing flexible electronics[1-4]. Wherein stretchable luminescent materials as vital components can provide emission sources and intrinsic flexibility to the optoelectronic displays for versatile applications in soft robots, intelligent sensing/detection, on-skin displays, and wireless communication (Fig. 1a)[5-9]. To date, most of the reported stretchable emitters are based on fluorescent polymers that only harness singlet excitons and suffer from short lifetimes[10-12]. Rare stretchable phosphorescent materials for advanced applications have been developed. Room-temperature phosphorescence (RTP) materials, featuring long emission lifetimes, tunable excited state properties, and high exciton utilization, have recently received increasing attention[13-16]. To obtain organic RTP materials, tremendous efforts have been devoted to promoting the intersystem crossing (ISC) process by incorporating heavy atoms, heteroatoms, and aromatic carbonyls into phosphors[17-20], and restraining

the non-radiative decay of triplet excitons by building a rigid environment[21,22]. Given the intrinsic merits of polymers in good processability, lightweight, and flexibility, polymer-based RTP materials become attractive alternatives to small molecules for expanding applications in stretchable photoelectronics[23-25]. Conventionally, RTP polymers can be obtained by chemically conjugating phosphors onto a polymer backbone via polymerization[26-28], or doping chromophores into rigid polymer hosts[29-32]. These homopolymers with rich hydrogen bonds and rigid microenvironments could suppress the non-radiative decay of phosphors for efficient RTP. However, the small free volume and high glass transition temperature ($T_g$) limit the movement and rearrangement of polymer chains, resulting in poor mechanical deformations. Generally, stretchable polymers require a low $T_g$ to meet the fast segmental dynamics at room temperature, which will cause intensive non-radiative deactivation and weaken RTP performance.

[1]Frontiers Science Center for Flexible Electronics (FSCFE), MIIT Key Laboratory of Flexible Electronics (KLoFE), Northwestern Polytechnical University, Xi'an 710072, China. [2]State Key Laboratory of Supramolecular Structure and Materials, Institute of Theoretical Chemistry, College of Chemistry, Jilin University, Changchun 130012, China. [3]Key Laboratory of Flexible Electronics (KLoFE) & Institute of Advanced Materials (IAM), Nanjing Tech University (NanjingTech), 30 South Puzhu Road, Nanjing 211816, China. [4]Research and Development Institute of Northwestern Polytechnical University in Shenzhen, Shenzhen 518057, China. ✉e-mail: iamlgu@nwpu.edu.cn; iamzfan@njtech.edu.cn; vc@nwpu.edu.cn

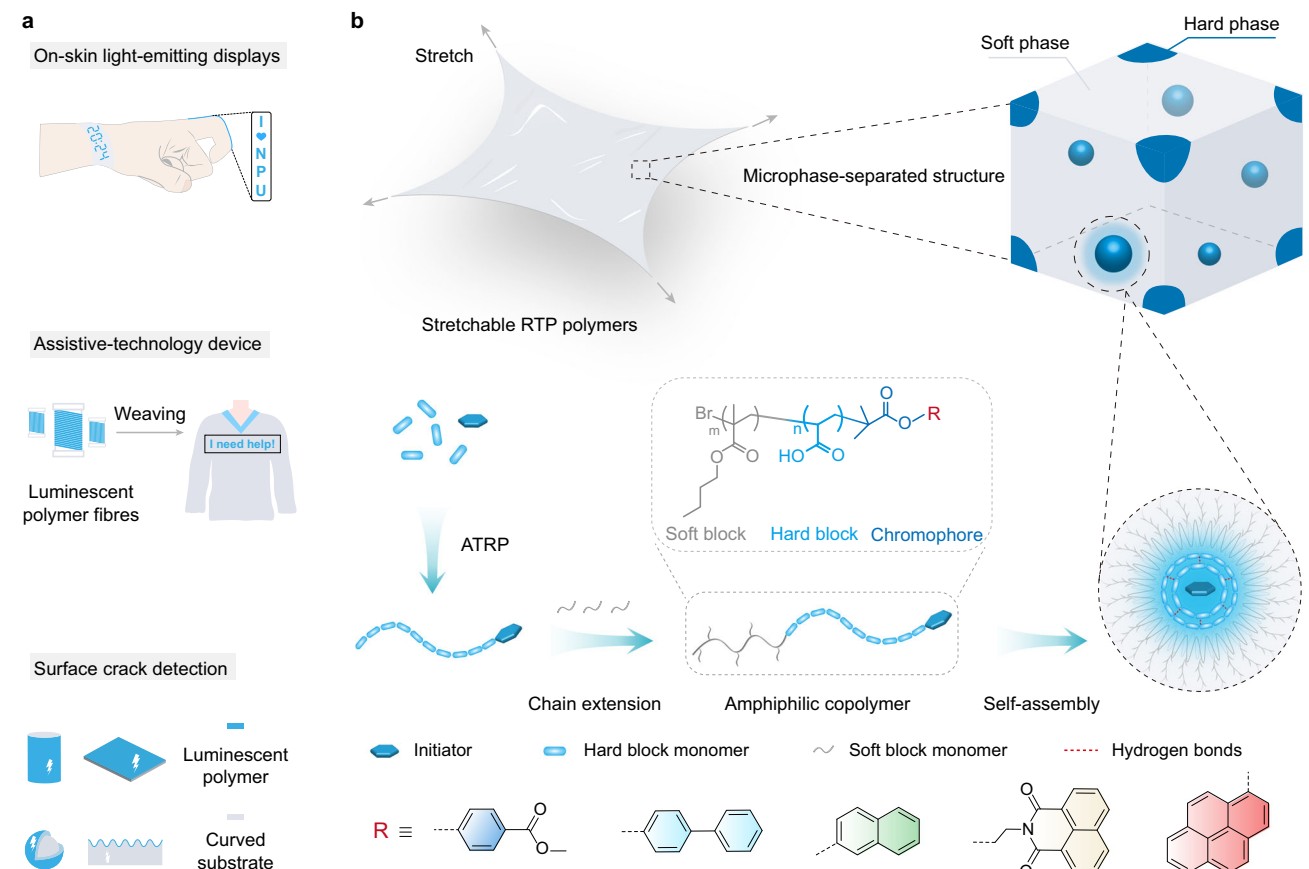

**Fig. 1 | Potential applications and design concept of stretchable phosphorescent materials. a** Potential application scenarios of stretchable luminescent materials in displays, sensing, and detection. **b** Rational design of stretchable room-temperature phosphorescence (RTP) polymers. Utilizing a two-step atom-transfer radical polymerization (ATRP) and subsequent hydrolysis, amphiphilic polyacrylic acid-co-poly (alkyl methacrylate) block copolymers are achieved. During hydrolysis, the block copolymer self-assembles into two-phase nanostructures. The polyacrylic acid hard phase with rich hydrogen bonding protects the phosphors for RTP generation, and the poly (butyl methacrylate) soft phase contributes to dissipating strain energy and thus enhancing stretchability. Meanwhile, stretchable copolymers with color-tunable RTP can be developed by rationally tailoring the initiator components and energy levels.

Therefore, achieving intrinsically stretchable polymers with long-lived phosphorescence remains a challenge.

Multiphase engineering could provide a facile platform to combine different components with multi-functions into a block copolymer[33–35]. Generally, block copolymers consist of two or more thermodynamic incompatible polymer segments linked by the covalent bond. The immiscibility of dissimilar segments generally induces a self-assembly process and forms periodic microphase-separated nanostructures[36–39]. These multiple microdomains can transfer or amplify their functions to the macroscopic scale of materials, thus giving rise to a portfolio of distinctly different properties in the same copolymer. Inspired by this, we reason that stretchable RTP materials might be obtained by multiphase engineering. This approach can efficiently combine stiffness and softness in the same polymer via programming the rigid microphase in the continuous soft domain of a block copolymer.

Here, we adopt atom-transfer radical polymerization (ATRP) to prepare block copolymers because of the precisely controlled molecular weight, composition, on-demand functionality, and wide utilization for diverse monomers[40]. A naphthalimide derivative is selected as ATRP initiator and phosphorescence chromophore because the multiple carbonyl groups and nitrogen heteroatom could effectively facilitate the population of triplet states through an efficient ISC process. Moreover, the rigid planar molecular skeletons can restrict molecular motions for suppressed non-radiative decay, contributing to RTP generation[41,42]. Considering the rich carboxyl groups can form strong hydrogen bonds to suppress the non-radiative transition of triplet excitons, we choose polyacrylic acid (PAA) as the hard block[24]. Meanwhile, soft monomeric units, including alkyl (meth)acrylates, carbosilane, siloxane, and ether chain, are widely used to decrease the $T_g$ for improving the flexibility or stretchability of polymer[43–45]. Particularly, alkyl methacrylate shows high ATRP activity, and the hydrophobic alkyl chains favor preventing phosphors from being quenched by surrounding moisture. Therefore, alkyl methacrylate monomers are introduced to prepare the soft blocks (Fig. 1b). After self-assembly of amphiphilic block copolymers, hard-soft microphase-separated structures can be formed. Wherein dispersed PAA hard phase provides a rigid environment to protect triplet excitons and thus promote the RTP generation. Continuous poly (alkyl methacrylate) phase with soft alkyl chains can enhance polymer chain dynamics, guaranteeing the stretchability of copolymers. Meanwhile, their mechanical performance can be well-tuned by varying the molecular structure and content of the soft blocks. Benefitting from multiphase engineering, stretchable RTP materials can be obtained.

## Results

### Materials synthesis and characterizations

Multiphase block copolymers were prepared by a two-step ATRP and subsequent hydrolyzation process (Supplementary Figs. 1–5). We synthesized bromoisobutyrate-modified naphthalimide (DBI) as the ATRP initiator. Then, macroinitiator PBM was synthesized with DBI, *tert*-butyl acrylate, CuBr₂/TPMA (catalyst), and Cu(0) mixtures under nitrogen at

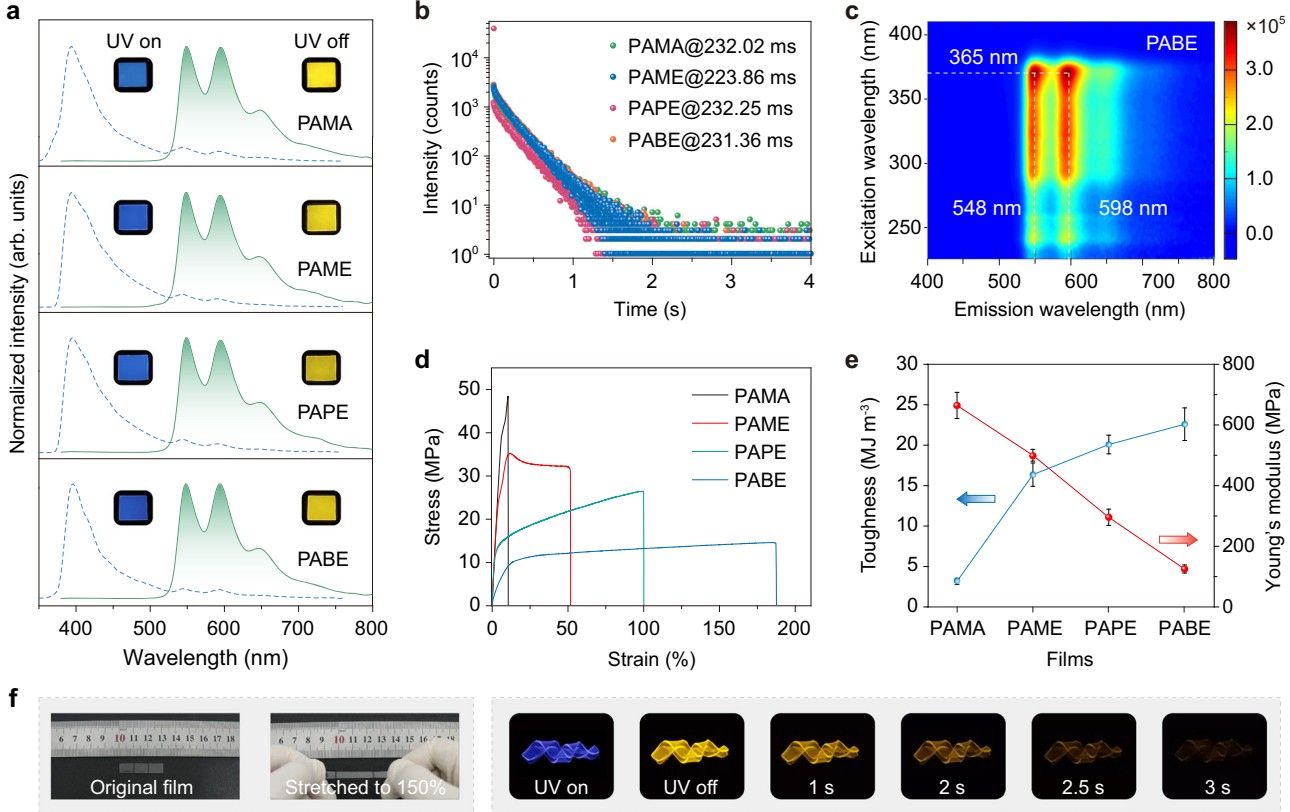

**Fig. 2 | Photophysical and mechanical properties of PAMA, PAME, PAPE, and PABE block copolymers. a** Normalized steady-state photoluminescence (blue lines) and phosphorescence (green lines) spectra of copolymers PAMA to PABE. Insets show corresponding photographs under 365 nm UV lamp on and off. **b** Lifetime decay curves of emission bands at 548 nm. **c** Excitation-phosphorescence mapping of PABE film under ambient conditions. **d** Typical stress-strain curves of polymer PAMA to PABE films, indicating tunability of the mechanical properties through rational molecular design. **e** Toughness and Young's modulus variation of PAMA to PABE films. Error bars represent mean ± standard deviation ($n = 3$). **f** Deformability and ultralong RTP demonstration of PABE film.

room temperature, followed by next chain extension utilizing PBM, alkyl methacrylate monomer, CuCl$_2$/PMDETA (catalyst), and stannous octoate in a given ratio. After the reaction finished, the mixture was precipitated in methanol/H$_2$O to give the as-prepared block copolymer, which was further dissolved in dichloromethane/trifluoroacetic acid mixture and hydrolyzed at room temperature for 24 h to give the final amphiphilic block copolymer. Considering the alkyl side chain lengths will influence $T_g$ and thus chain dynamics of corresponding copolymers, we selected methyl methacrylate (MMA), ethyl methacrylate (EMA), propyl methacrylate (PMA), and butyl methacrylate (BMA) as the second block monomers for preparing amphiphilic block copolymers PAMA, PAME, PAPE, and PABE, respectively. The chemical structures and purity of ATRP initiators were fully confirmed by nuclear magnetic resonance ($^1$H and $^{13}$C NMR) spectroscopy, high-performance liquid chromatography-mass spectrometry (HPLC-MS), and elemental analyses (Supplementary Figs. 6–34). As demonstrated by Fourier transform infrared spectroscopy (FTIR) spectra, the hydroxy stretch at around 3100–3600 cm$^{-1}$ indicated that the *tert*-butyl ester groups were converted into carboxyl moieties after hydrolyzation, and the shift of carbonyl stretching resonance verified the formation of the C=O•••H–O hydrogen bonds (Supplementary Fig. 35). The resulting polymer molecular weight and polydispersity (PD) were determined by gel permeation chromatography (GPC) (Supplementary Figs. 36–39 and Supplementary Tables 1–3).

## Photophysical and mechanical studies
To understand the influence of alkyl side chain lengths of the soft block on the copolymer performance, we investigated the photophysical

properties of PAMA, PAME, PAPE, and PABE containing equimolar acrylic acid (AA) and soft block monomers. Under 365 nm excitation, all films displayed blue emission colors with photoluminescence (PL) bands at 395 nm (Fig. 2a and Supplementary Fig. 40). After stopping the ultraviolet (UV) lamp excitation, intense yellow afterglow lasting for 3 s can be observed (Supplementary Fig. 41 and Supplementary Movie 1). With a delay time of 8 ms, PAMA to PABE showed identical spectra with prominent peaks at 548 and 598 nm. Meanwhile, their lifetime decay profiles at 548 nm exhibited almost equal lifetimes up to 232.25 ms, indicating the phosphorescence characteristic (Fig. 2b and Supplementary Table 4). Additionally, phosphorescence spectra remain constant emission bands under varied excitation wavelengths, suggesting stable triplet emissive centers deriving from DBI units in these multiphase systems (Fig. 2c and Supplementary Fig. 42). These results are consistent with control polymers HPAA and NPAA, indicating the incorporation of soft segments with different species from MMA to BMA as well as the terminal bromine atom at the polymer chain had little impact on the photophysical properties of these block copolymers, but significantly affected their mechanical properties (Supplementary Figs. 43, 44 and Supplementary Tables 5, 6). As shown in Fig. 2d and e, compared with homopolymer HPAA, the strain-at-break of block copolymers had a dramatic enhancement and increased gradually as increasing alkyl side chain length of the soft blocks, with a maximum of 188% for PABE film (Supplementary Fig. 45). The calculated toughness, related to energy dissipation, showed the highest value of 23.6 MJ m$^{-3}$ in PABE film (Supplementary Fig. 46 and Supplementary Table 7). In contrast, fracture strength and Young's modulus decreased from PAMA to PABE films, consistent with the varying trend

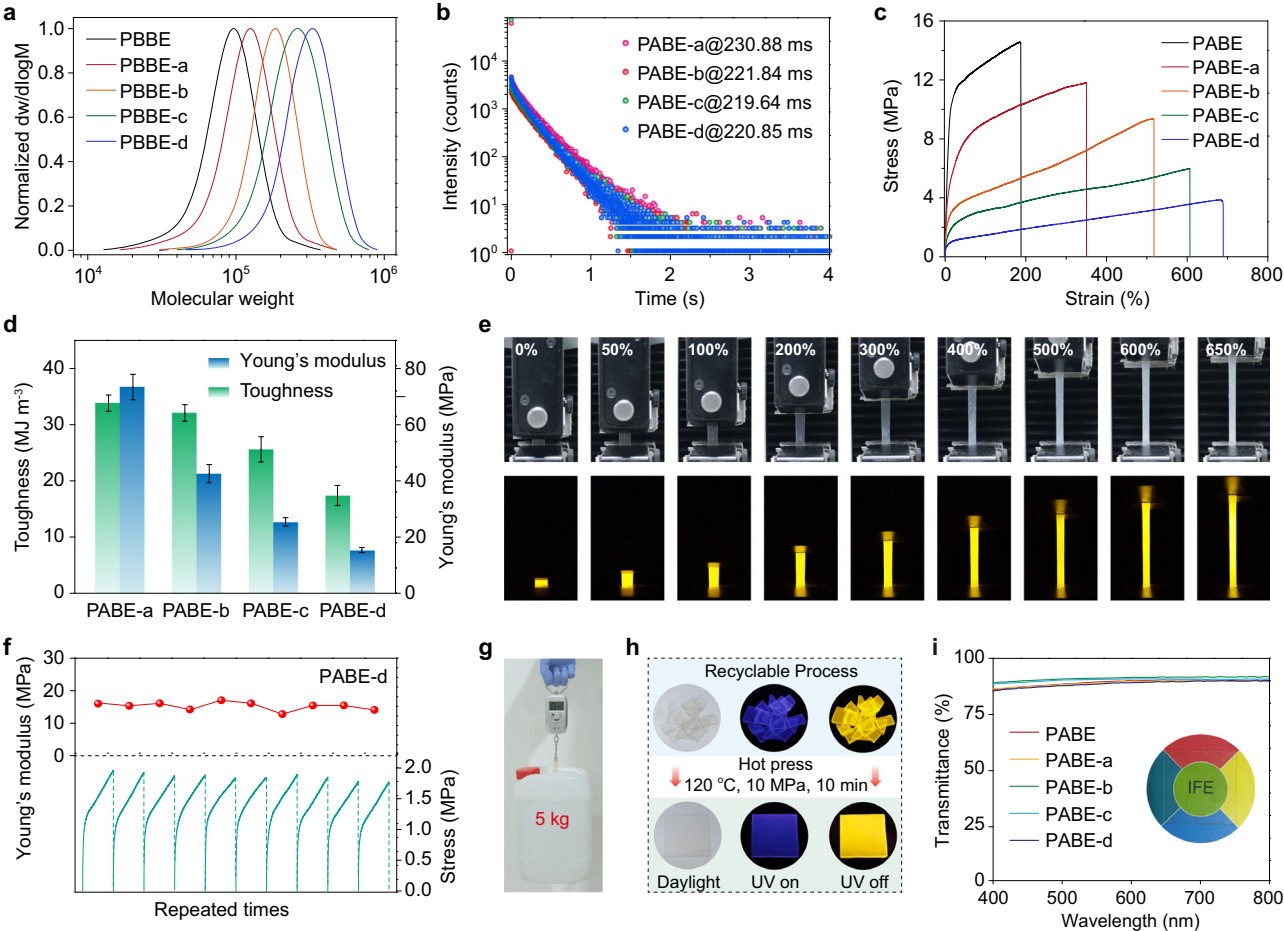

**Fig. 3 | Photophysical and mechanical properties of PABEs with varying BMA mass fraction. a** GPC traces of the unhydrolyzed copolymers PBBE to PBBE-d. The molecular weight ($M_n$) can further determine the content of AA and BMA in PABE to PABE-d. According to the living character of ATRP, the molar ratio of AA/BMA was 1:1, 1:3, 1:5, 1:7, and 1:9, respectively. **b** Lifetime decay curves of emission bands at 548 nm. **c** Stress-strain curves of PABE to PABE-d films. **d** Calculated toughness and Young's modulus of copolymers PABE-a to PABE-d. Error bars represent mean ± standard deviation ($n = 3$). **e** Real-time photographs of PABE-d film during stretching on a universal testing machine taken under daylight and after stopping the 365 nm excitation. **f** Repeated tensile tests of PABE-d film with a 200% strain 20 times (each repeated test needs some recovery times for the sample). The green lines show the stress-strain curves at the 2nd, 4th, 6th to 20th tests, and the red balls represent related Young's modulus. **g** Photograph of PABE-d film when lifted with heavy loads. **h** Photographs of the cut PABE-d film pieces and after hot pressing (120 °C, 10 MPa, 10 min), demonstrating its good recyclability. **i** Optical transmittance of various PABEs films in the visible range. Inset: photograph of PABE film (thickness: 0.4 mm).

of their $T_g$, demonstrating lower stiffness and easier deformation for PABE film (Supplementary Figs. 47–49). Therefore, stretchable and flexible RTP polymer can be achieved by synthesizing amphiphilic block copolymers (Fig. 2f). Moreover, increasing alkyl side chain lengths of the soft blocks can effectively enhance polymer chain dynamics and improve stretchability.

To further optimize the mechanical properties of the copolymer, we synthesized PABE-a to PABE-d with varied molecular weights of the soft blocks (Fig. 3a and Supplementary Table 2). As shown in Fig. 3b, as the molecular weight increased gradually, the photophysical properties of PABE-a to PABE-d films were almost unchanged, maintaining long-lived phosphorescence lifetimes over 220 ms (Supplementary Figs. 50–52 and Supplementary Tables 8, 9). In contrast, the mechanical performance obviously depended on the BMA content. As the molar ratio of AA/BMA changed from 1/3 to 1/9, the stretchability of the copolymer films significantly improved from 350% to 690%, far exceeding the reported homogeneous RTP polymer films, and PABE-d film could keep an elongation of beyond 600% when the tensile rate was below 50 mm min⁻¹ (Fig. 3c, Supplementary Figs. 53, 54, and Supplementary Table 10). Meanwhile, as relative proportion of acrylic acid decreased, the toughness of PABE to PABE-d films decreased from 33.8

to 17.4 MJ m⁻³, and Young's modulus reduced from 112.8 to 15.4 MPa (Fig. 3d and Supplementary Table 11). From these experiments, we found that varying molecule weights of the PBMA block can significantly regulate the mechanical properties of copolymers but almost not affect the ultralong phosphorescence. In agreement with our multiphase design concept, the PAA hard block not only provides the rigid microenvironment for generating RTP but also serves as the physical cross-linking point to modulate the stiffness of copolymers. While the PBMA soft block mainly determines the elongation ability of PABEs samples.

Given the decent performance of PABE-d film, we next investigated the optical and mechanical stabilities. As shown in Fig. 3e, PABE-d film can maintain bright ultralong RTP under different stress and strain conditions (Supplementary Figs. 55, 56 and Supplementary Movie 2). Moreover, it can be repeatedly stretched to a 200% strain 20 times with stable Young's modulus, after which it can still hold a weight of 5 kg (Fig. 3f and g). These results suggested their good mechanical durability and optical stability under large-strain states. Furthermore, benefitting from the high thermal stability and softness of the PBMA segment, PABE-d film performed good recyclability via hot pressing (Fig. 3h and Supplementary Figs. 57, 58). Therefore, our ingenious

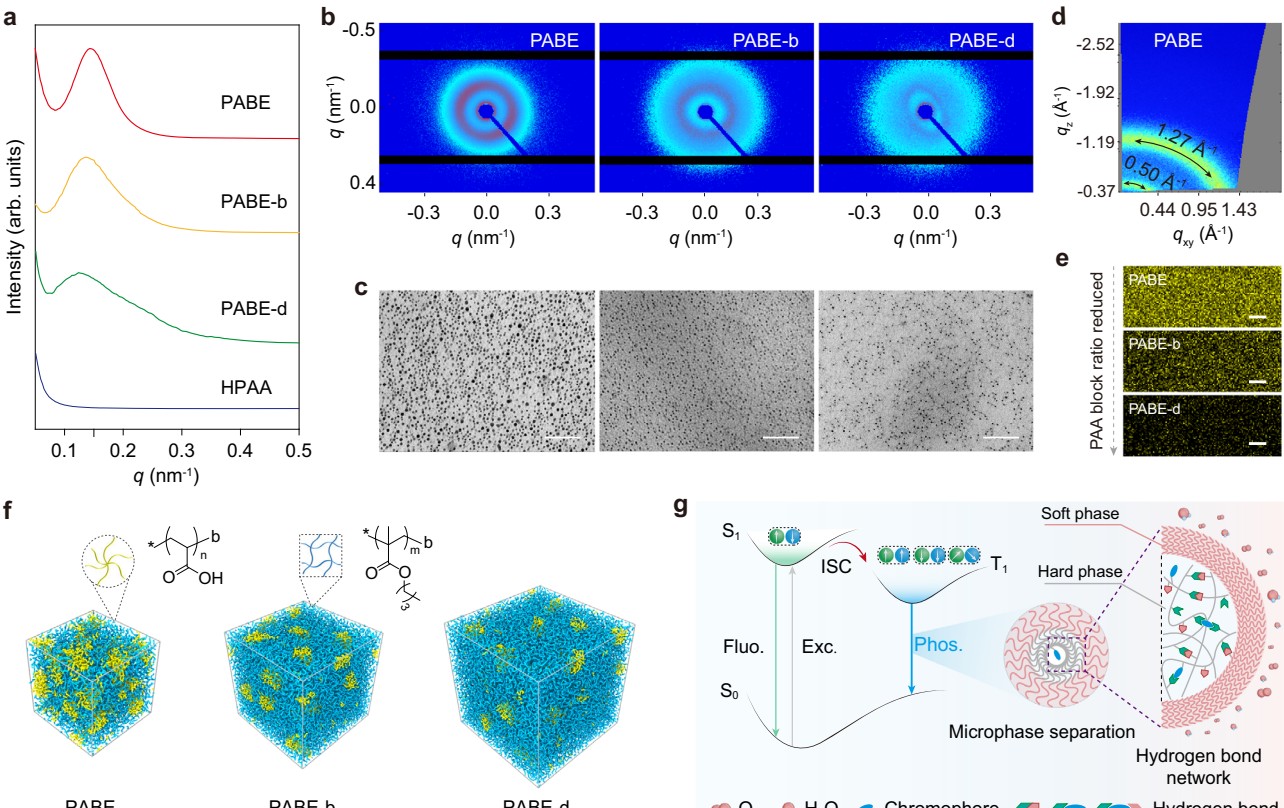

**Fig. 4 | Mechanism of stretchable polymers with ultralong phosphorescence.** **a** SAXS profile plots of copolymers PABEs and homopolymer HPAA, supporting the microphase-separated morphology in the designed block copolymers, while HPAA shows no scattering peak. **b** Related 2D SAXS images of PABEs films. **c** TEM images of PABE (left), PABE-b (middle), and PABE-d (right) provide clear evidence of the microphase-separated structure with PAA phase (dark spheres with diameters of 10–30 nm) dispersed in continuous PBMA phase (gray areas). The PAA segment was selectively stained with uranyl acetate. Scale bar: 300 nm. **d** 2D-WAXS pattern of

PABE film. **e** Confocal fluorescence images of PABE, PABE-b, and PABE-d. $\lambda_{ex} = 405$ nm. Scale bar: 20 μm. Yellow channel (570–600 nm): phosphorescence of DBI in PAA matrix. **f** Snapshots of the simulated structures for PABE, PABE-b, and PABE-d in the bulk. These amphiphilic copolymers spontaneously assemble into microphase-separated structures. **g** Proposed mechanism of ultralong phosphorescence from amphiphilic block copolymers. Fluo., Phos., and Exc. refer to fluorescence, phosphorescence, and excitation, respectively.

multiphase design endows block copolymers with integrated advantages in good mechanical properties, optical and thermal stability, recyclability, water resistance, and ultralong phosphorescence, providing a facile strategy for obtaining stretchable RTP materials (Supplementary Figs. 59, 60). Additionally, despite our designed copolymers consisting of immiscible PAA and PBMA segments, they exhibit high optical transparency of up to 90% in the visible region, indicating good uniformity and no macroscopic phase separation of these polymers (Fig. 3i). Based on these data, we speculate that there might form microphase-separated structures in these block copolymers contributing to their decent stretchability and RTP performance.

### Mechanism of stretchable phosphorescent copolymers

To confirm our speculation, we further systematically investigated the microstructures of various PABEs films. As shown in Fig. 4a, small-angle X-ray scattering (SAXS) profile plots displayed broad and intense scattering peaks for PABE, PABE-b, and PABE-d films, whereas no obvious scattering peaks can be detected from polymers PBBE and HPAA (Supplementary Fig. 61). This result indicated the formation of multiple microphases in PABEs copolymer films. Moreover, from PABE to PABE-d, the scattering peaks gradually shifted to the low $q$ region along with decreased intensity, indicating increased inter-domain spacings ($d = 2\pi/q$) and less prominent accumulation of the hard segments, thus reducing Young's modulus. The 2D SAXS images showed that electron density contrast between the two domains decreased as PAA contents reduced, proving that the immiscibility of soft-hard

blocks, as well as hydrogen bonds between PAA chains, could provide the driving force for microphase separation (Fig. 4b)[46,47]. The microphase-separated nanostructures were further certified by transmission electron microscopy (TEM) utilizing ultra-thin section specimen (Fig. 4c). The TEM micrographs reveal clear two-phase morphology, wherein spherical PAA hard phases were uniformly dispersed in continuous PBMA soft phase. As the ratio of AA/BMA varied from 1/1 to 1/9, the soft domains increased with larger inter-domain spacings, in accordance with the SAXS results. This variation presumably contributes to the different stretchability of PABEs films. Additionally, all PABEs films showed $T_g$ at around 25 °C, providing a prerequisite for chain segmental motion to achieve flexibility and stretchability of copolymers at room temperature. Importantly, this low $T_g$ was consistent with that of homopolymer PBMA, indicating the stretchability of resulting copolymers mainly derived from the soft PBMA block (Supplementary Fig. 62). These data agree with the multiphase design concept, wherein the soft microphase provides fast segmental dynamics for good stretchability of copolymers.

To further reveal the origination of ultralong RTP, we conduct wide-angle X-ray scattering (WAXS) experiments of various copolymers. As shown in Fig. 4d, only two broad scattering bands at 0.50 Å$^{-1}$ (7.11°) and 1.27 Å$^{-1}$ (18.00°) ascribed to the characteristic bands from PBMA can be observed, without other aggregates-induced π-π interactions (Supplementary Figs. 63–67). Combining the photophysical properties of DBI in doped-PAA film and dilute solution ($10^{-5}$ M) at 77 K, we infer that ultralong RTP at 548 nm of PABEs copolymers derived

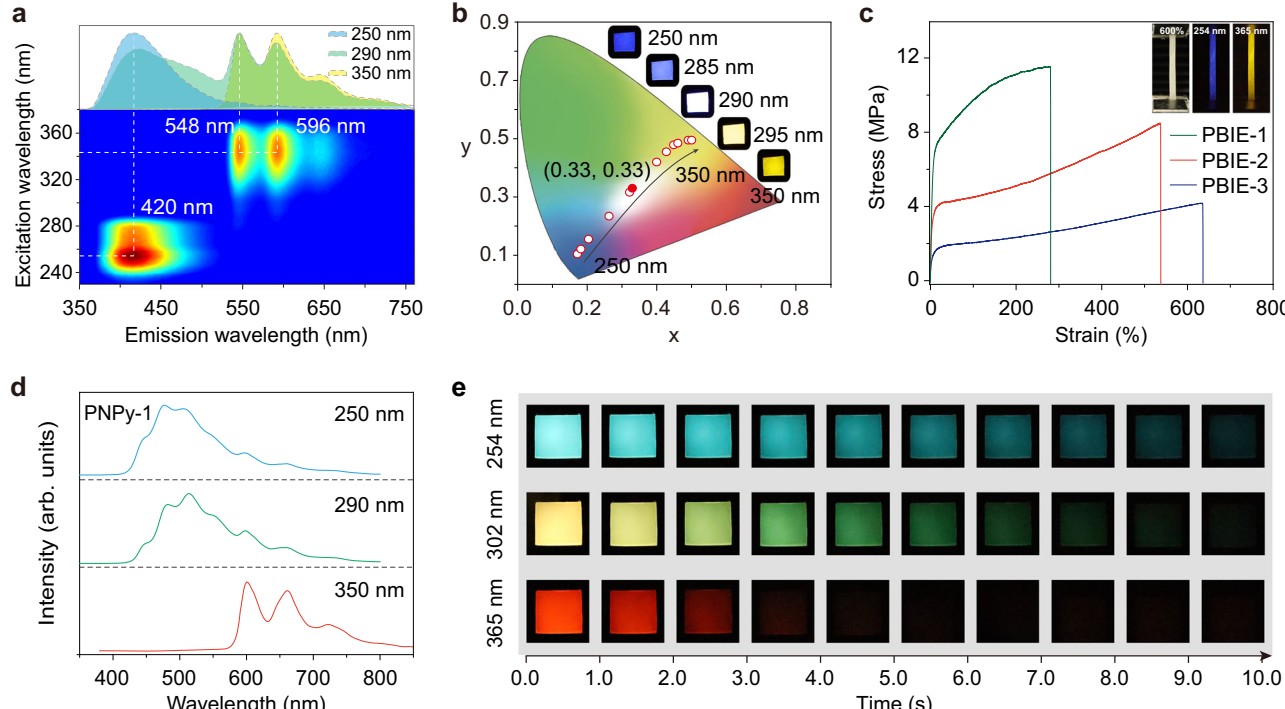

**Fig. 5 | Stretchable polymers with color-tunable ultralong phosphorescence.**
**a** Excitation-phosphorescence mapping and phosphorescence spectra of PBIE-1
film excited by 250, 290, and 350 nm. **b** CIE chromaticity diagram for copolymer
PBIE-1 with excitation ranging from 250 to 350 nm. Inset: long-lived phosphores-
cence photographs of PBIE-1 excited at 250, 285, 290, 295 and 350 nm, respectively.
**c** Stress-strain curves of PBIE-1 to PBIE-3 films. Inserts: photographs of PBIE-3 film
with a 600% strain taken under daylight and after stopping the 254 and 365 nm
excitation. **d** Phosphorescence spectra of PNPy-1 film excited by 250, 290, and
350 nm. **e** Excitation- and time-dependent phosphorescence photographs of PNPy-
1 taken after ceasing the 254, 302, and 365 nm irradiation.

from the isolate molecule phosphorescence of DBI chromophore in
the hard PAA phase (Supplementary Figs. 68–73). Meanwhile, we uti-
lized confocal fluorescence microscopy to trace the RTP origination in
our multiphase systems (Fig. 4e). Under 405 nm laser irradiation, the
collected yellow channel imaging demonstrated that the emitting
centers were uniformly dispersed in films. As the content of the hard
block decreased, the emission areas became sparse, which accorded
with the variation tendency of the PAA microphase distribution in the
TEM images. This result indicates that the ultralong phosphorescence
of these copolymers mainly originated from the hard PAA microphase,
which can confine phosphors via multiple hydrogen bonds. Additional
control experiments can also verify this description. Due to the lack of
strong hydrogen bonds, almost no RTP was observed from control
polymers PBM, PBBE-a, and PBMA under ambient conditions (Sup-
plementary Figs. 74–77). Therefore, we conclude that the hard and soft
phases based on multiphase engineering are indispensable for
achieving stretchable RTP polymers.

To strengthen our conclusions, large-scale coarse-grained mole-
cular dynamics simulations were performed to reveal the self-assembly
behavior and multiple microphase formation of these block copoly-
mers (Supplementary Tables 12, 13, Supplementary Data 1). As shown
in Fig. 4f, after $1.2 \times 10^6$ τ simulation, microphase-separated structures
of amphiphilic block copolymers PABE, PABE-b, and PABE-d in the bulk
are obtained. With increasing the volume fractions of the soft block,
the PAA microphase was widely distributed in the PBMA domain with
larger inter-domain spacings (Supplementary Fig. 78). These theore-
tical simulation results agree well with the SAXS and TEM data, illus-
trating the formation of hard-soft microphase-separated structures in
these block copolymers.

Taken together, we proposed a reasonable mechanism for
stretchable RTP copolymers. As shown in Fig. 4g, benefitting from the
multiphase design, amphiphilic block copolymers self-assemble into
microphase-separated structures, wherein dispersed hard phases are

uniformly embedded in the continuous soft phase. The hard micro-
phase contains chromophores and PAA chains. Under the UV lamp
excitation, triplet excitons can be populated through the ISC process
(Supplementary Fig. 79, Supplementary Data 2). Due to the multiple
hydrogen bonds between polymer chains and chromophores, the hard
phase provides a rigid microenvironment to confine molecular
motions of chromophores and decrease the quenching from sur-
rounding oxygen. Therefore, the triplet state excitons are stabilized
and non-radiative transition is effectively suppressed for generating
ultralong RTP (Supplementary Fig. 75). Meanwhile, the PBMA soft
phase with a low $T_g$ plays a crucial role in enhancing the polymer chain
dynamics, thereby rendering a decent stretchability for the block
copolymers (Supplementary Fig. 80). Besides, the tangled hydro-
phobic soft segments provide an extra barrier for the triplet excitons
and further prevent the quenching from surrounding moisture. Under
the synergism of hard and soft microdomains, copolymers featuring
ultralong RTP, decent mechanical properties, thermal and optical
stability, and water resistance can be achieved.

## The universality of the multiphase engineering
To establish the generality of multiphase engineering, stretchable
copolymers with color-tunable RTP were further developed by
rationally tailoring the initiator components and energy levels (Sup-
plementary Fig. 79). We expand the binary initiator system PBIEs
(Supplementary Figs. 81–84). As excitation wavelengths varied from
230 to 380 nm, the RTP emission center showed a dramatic bath-
ochromic shift from 420 to 548 nm (Fig. 5a and Supplementary
Table 14). Notably, under 290 nm excitation, pure white phosphores-
cence with CIE coordinate of (0.33, 0.33) can be achieved, along with
tunable mechanical properties (Fig. 5b, c, and Supplementary
Table 15). Meanwhile, the ternary initiator system PNPys can further
expand the RTP color tunability ranging from 476 to 600 nm with
ultralong lifetimes of 981.11, 863.02, and 244.06 ms, respectively

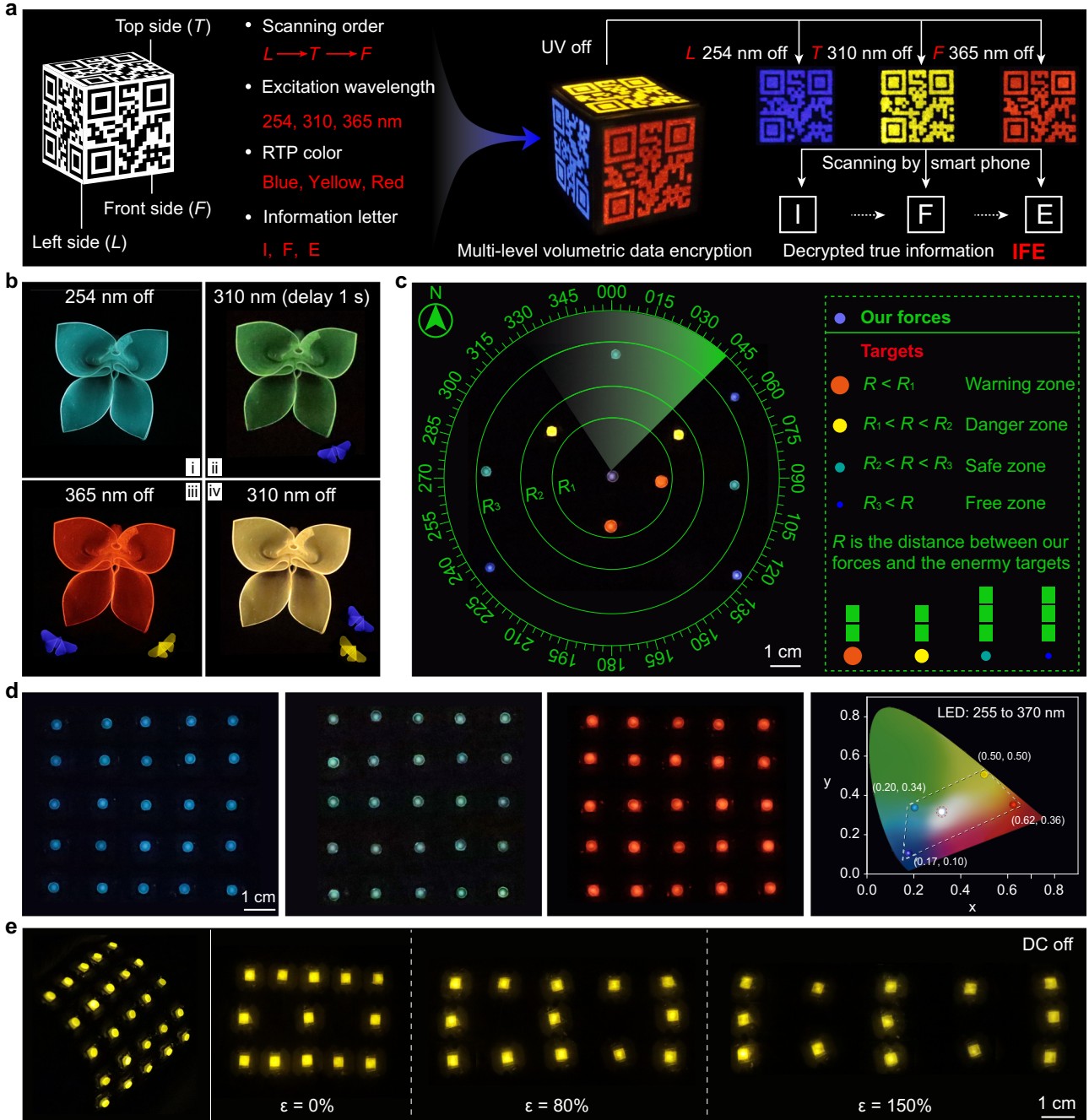

**Fig. 6 | Demonstration of stretchable phosphorescent polymers for multi-level data encryption and multicolor afterglow display. a** Schematic illustration of multi-level volumetric data encryption utilizing PBIE-3 (L and T sides) and PNPy-3 (F side) samples, and photographs of the QR codes on a cube after switching off different UV lamps and the corresponding screenshot of the identified letters I, F, and E. **b** Photographs of the hand-folded flower (PNPy-3 film) and butterfly (PBIE-3 film) taken after stopping different UV light excitations. **c** Demonstration of radar detection mimicry by controlling direct current (DC) off. **d** RGB afterglow display arrays based on PNPy-3 films under DC switched off. The CIE chromaticity diagram shows the wide gamut coverage of PNPy-3 and PBIE-3 films under different LED light excitations, demonstrating the potential of color-tunable copolymers in full-color displays. **e** Demonstration of the display arrays subjected to bending and stretching to different strains under DC switched off.

(Fig. 5d, e, Supplementary Figs. 85–89, and Supplementary Tables 16, 17). Notably, PNPy-3 film displayed a maximum elongation of up to 712%. Besides, stretchable copolymer films with efficient RTP can also be obtained by utilizing Br-modified phosphors (Supplementary Figs. 90, 91 and Supplementary Table 18).

## Potential applications
Combining the color-tunable RTP and stretchability of these block copolymers, we explored their potential in data encryption and flexible multicolor afterglow display. As shown in Fig. 6a, based on screen printing, three quick response (QR) codes carrying I, F, and E information were painted on different sides of a cube. Only when we obey the right excitation wavelength, RTP color, and scanning order can we get the true information IFE, demonstrating the potential of our block copolymers in multi-level volumetric information encryption (Supplementary Fig. 92, Supplementary Movie 3). Meanwhile, the flexible copolymer films can be made into different handicrafts for information storage. Due to the dynamic afterglow of PNPy-3 and

PBIE-3 films, the variation of a flower from budding in spring to withering in autumn, accompanied by the butterflies attracted and then flying away was recorded, which can be potentially applied in instant information recording and reading (Fig. 6b, Supplementary Fig. 93). Besides, a multicolor afterglow display can be realized in one copolymer film by controlling electrical excitation (Supplementary Figs. 94–103, Supplementary Table 19). Color-tunable RTP copolymer films were utilized as display screens and circuit diagram was introduced to fabricate the photoelectric device for afterglow display. As shown in Fig. 6c, a demo of radar all-round detection can be realized by controlling direct current (DC) off (Supplementary Movie 4). The colorful afterglow represents the targets in different ranges, demonstrating the potential of these copolymers for accurate positioning and warning. Furthermore, the stretchable copolymer films can also be utilized for large-area full-color afterglow display (5 × 5-pixel array), while maintaining their resistance to large mechanical deformation, including bending and stretching up to 150% strain (Fig. 6d, e, and Supplementary Movie 5).

## Discussion

In conclusion, we have presented a multiphase design strategy for stretchable phosphorescent materials that combine stiffness and softness into an amphiphilic block copolymer. The well-defined design and ATRP synthesis offer versatility in tuning polymer structures and mechanical properties. Impressively, the maximum tensile strain could reach up to 712%, maintaining an ultralong phosphorescence lifetime of 981 ms. The ultralong RTP could be observed even under repeated strain, demonstrating good mechanical durability and optical stability. The long-lived RTP was ascribed to the formation of microphase-separated nanostructures in copolymers. Besides, multiphase engineering could generally apply to the binary and ternary initiator systems for realizing color-tunable phosphorescence in the visible range. We also demonstrated their potential in multi-level volumetric data encryption and stretchable multicolor afterglow display combining stretchability and color-tunable phosphorescence. This work will provide a platform for designing intrinsically stretchable phosphorescent materials and extend the potential of phosphorescence polymers.

## Methods

### Reagents and materials

*tert*-Butyl acrylate (TBA, 99%), cupric bromide (CuBr$_2$, 99.9%), tris(2-pyridylmethyl)amine (TPMA, 98%), *N,N*-dimethylformamide (DMF, 99.8%), methanol (MeOH, 99.9%), butyl methacrylate (BMA, 99%), cupric chloride (CuCl$_2$, 98%), pentamethyldiethylenetriamine (PMDETA, 98%), anisole (99%), stannous octoate (95%), tetrahydrofuran (THF, 99.5%), dichloromethane (DCM, 99.5%), trifluoroacetic acid (TFA, 99.5%), methylparaben (99%), triethylamine (99.5%), 2-bromoisobutyryl bromide (98%), 4-phenylphenol (99%), 2-naphthol (99%), 1,8-naphthalic anhydride (99%), ethanolamine (99%), ethanol (99.9%), and 1-hydroxypyrene (98%) were purchased from commercial sources without further purification unless otherwise stated. All the purchased monomers were purified by passing through a column of basic alumina to remove inhibitors. For flash column chromatography, silica gel with 200–300 mesh was used.

### Synthesis of macroinitiator

**PBM**. Initiator DBI (0.10 g, 0.256 mmol), TBA (9.85 g, 76.88 mmol), CuBr$_2$ (1.71 mg, 7.68 μmol), and TPMA (8.92 mg, 0.031 mmol) were dissolved in DMF (4 mL) and degassed by purging with nitrogen for 30 min. Polymerization commenced upon the addition of a piece of Cu(0) to the degassed reaction mixture. The solution was stirred at room temperature for 12 h and then precipitated in MeOH/H$_2$O mixtures to give the macroinitiator PBM. $M_n$: 27160; PD: 1.11.

### Synthesis of block copolymer

**PABE**. Macroinitiator PBM (1.5 g, 0.057 mmol), BMA (3.22 g, 22.67 mmol), CuCl$_2$ (4.60 mg, 0.034 mmol), and PMDETA (11.85 mg, 0.068 mmol) were dissolved in anisole (4 mL). The mixture was degassed by purging with nitrogen for 20 min. After adding degassed stannous octoate (13.86 mg, 0.034 mmol), the mixture was stirred at 70 °C for 20 h. Upon cooling, the catalyst was removed by passing a solution of the polymer in THF over a neutral alumina plug, and then the filtered polymer solution was precipitated in MeOH/H$_2$O mixture to give the as-prepared PBBE block copolymer ($M_n$: 68800; PD: 1.24). Finally, polymer PBBE was dissolved in DCM and 10 mL TFA was slowly added to the mixture. After stirring at room temperature for 24 h, the solution was removed by rotary evaporation to yield amphiphilic block copolymer PABE film. Following the same procedure, PABE-a, PABE-b, PABE-c, and PABE-d block copolymers were prepared by varying the molar feed ratio of PBM/BMA to 1/600, 1/1000, 1/1500, and 1/2000, respectively. To obtain regular test specimens, the above polymer film was further hot pressed in a mold at 120 °C.

## Data availability

The authors declare that all the data supporting the findings of this study are provided in the manuscript and its supplementary information files, or available from the corresponding authors on request. Source data are provided with this paper.

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

## Acknowledgements

This research is supported by the National Natural Science Foundation of China (62288102 (W.H.), 52203242 (L.G.), and 62134007 (W.H.)), National Key R&D Program of China (grant no. 2020YFA0709900 (W.H.)), the Natural Science Foundation of Shaanxi Province (2022JQ-581 (L.G.)), the Natural Science Foundation of Ningbo (2021J054 (L.G.)), the Guangdong Basic and Applied Basic Research Foundation (2021A1515110367 (L.G.)), Zhejiang Provincial Natural Science Foundation of China under Grant No. LQ23B020004 (L.G.). We are grateful to the Analytical & Testing Center of Northwestern Polytechnical University for supporting the test resources.

## Author contributions

L.G., Z.A., and W.H. conceived the experiments. N.G., Z.A. and L.G. wrote the manuscript. N. G. and X.Z. conducted the experiments. L.W. helped the characterizations of molecular structures. X.Z. helped to take photos and edit the movies. Z.Q., H.Q., A.L. and H.M. contributed to theoretical calculations. All authors contributed to the data analyses.

## Competing interests

The authors declare no competing interests.
