## [Peer Review File · Nature Communications]

Stretchable phosphorescent polymers by multiphase engineeringEditorial Note: Parts of this Peer Review File have been redacted as indicated to remove third-party material where no permission to publish could be obtained.

REVIEWER COMMENTS

Reviewer #1 (Remarks to the Author):

This work reported a kind of block copolymer based on multiphase engineering. The as-prepared polymer showed both rigidity and flexibility, and a room temperature phosphorescent material with long-lived emission and high stretchability was acquired. In addition, the authors demonstrated the application of this material in multi-level volumetric data encryption and stretchable afterglow display. This is an interesting work with sufficient data. I recommend its publication after revisions, and there are some confusions need to be solved.

1. How many times are the measurements for the stress-strain curves? I recommend that at least three times parallel measurements are implemented for the curves in Fig. 2, and the error bar should be provided. Moreover, what is the detailed measurement information for the mechanical measurements?
2. The authors have mentioned the good water resistance for the copolymer. What are the performances for these copolymers in water? Would the water affect the hydrogen bonds and thus the RTP performances?
3. Why selected the naphthalimide derivative as the chromophores in this work?
4. The authors have demonstrated the function of hydrogen bonds in enhancing the RTP emissions. Are there any referenced experiments to validate this demonstration?
5. To prepare the films, the copolymers were hot-pressed under 120 °C. Would this process affect the molecular arrangement or RTP performances of the materials?

Reviewer #2 (Remarks to the Author):

This manuscript introduced stretchable block copolymers with ultralong room temperature phosphorescence. It achieved a good stretchability of 712% and an ultralong phosphorescence lifetime of up to 981.11 ms. Although the author describes the difficulty of obtaining long afterglow and stretchable properties, there are still some relevant reports. Unfortunately, the RTP quantum yield here is not as good as these reports due to the phosphors used in this manuscript are very common. This manuscript does not demonstrate the importance of excellent stretchability in applications. In addition, the author spends a lot of time explaining the mechanism of stretchable phosphorescent copolymers. They said it can efficiently address the dilemma between the “rigidity” and “flexibility” in the same polymer. But the good stretchability requires sacrificing RTP QYs. The QYs were significantly decreased from 2.2% (PABE) to 0.97%(PABE-d). A similar work using only PAA can reach a phosphorescence

quantum yields of 4.0% (Ref: Journal of Luminescence 263 (2023) 119978). And many stretchable RTP polymer have been reported previously (DOI: 10.1021/jacs.2c13264, 10.1002/adom.202102449). I do not recommend this manuscript accepted by such a high-level journal as Nature Communication because this work lacks importance and innovation.

Questions:

- (1) The authors observed the increased distance between adjacent emitters through confocal fluorescence microscopy. But the distance between adjacent emitters is at the nanoscale by TEM, which far exceeds the resolution of a fluorescence microscope. It cannot “matched the distribution of the PAA microphase in the TEM images”.
- (2) The final polymer was precipitated in MeOH/H₂O mixtures. Why the hydrophilic PAA is located inside the self-assembled structure, while hydrophobic PBM is located outside?
- (3) Does TGA curves indicate solvent residue in the polymer ?
- (4) Does the phosphor will be brominated by the bromine radical, which is generated by the ATRP process?
- (5) How does the binary initiator system work? How about the simple mixture of afterglow?
- (6) What’s the ratio of the two phosphors in the binary initiator system? It’s unlikely to be the feed ratio.
- (7) The impurity shows great influence on the phosphorescence, especially afterglow (10.1038/s41563-020-0797-2). The origin of the afterglow should be further investigated. A strict purification was suggested for the phosphor-based initiators. Then, the phosphor-based initiators are suggested to dope into polyacrylic acid physically to testify whether the afterglow is generated by the initiator.

Reviewer #3 (Remarks to the Author):

Journal: **Nature Communications**

Title: **Stretchable phosphorescent polymers by multiphase engineering**

Authors have successfully demonstrated the fabrication of stretchable block copolymer based phosphorescent materials. Importantly, the as-proposed synthetic methodology provides convenient optimization of the as-proposed block system with tunable properties. The research carried out seems to have significant impact and the proposed methodology is convincing. Although, I request authors to address the following comments, thereby suggesting the major revision for the present stage.

I suggest authors to address the following comments,

1. Authors have emphasized the significance of framing the stretchable room temperature phosphorescent materials, but failed to stress the choice of selection of chromophore units. How do the as-presented chromophore materials fit in the desired synthetic methodology?

2. The discussion part with respect to synthesis and its corresponding characterization is limited. Authors should impart some considerable portion with respect to synthetic portions.
3. Why ATRP method is specifically chosen? What are the challenges in other methods? What are the specific limitations in ATRP synthesis?
4. The reason for selecting the hard phase PAA is vague. How do authors believe that PAA phase can prevent the triplet exciton quenching behavior?
5. Alkyl methacrylate soft phase is chosen, apart from this, there are many other soft monomeric units. Authors can mention various soft monomers and state the key reason for selecting the specific soft phase in the introduction part.
6. Supplementary table 1 results reveals that upon increasing the chain length, the resultant polymer's molecular weight is increased. Interestingly, the PDI values experiences some irregular trends, why it is so?
7. From Supplementary table 4, it is evident that PABE shows the longer average lifetime value, but why PABE is considered for further studies?
8. The quantum yield of the as-obtained coblock systems are within the range of 1.2 to 2.2, Why the as-obtained quantum yields results are lower? Authors can attribute the reason for such lower quantum yield values. It is highly valued to know the control chromophore quantum yield values, so that audience can understand the significance of the as-obtained coblock system.
9. The glass-transition values and its reduction upon increasing the chain length is interesting. Authors can attribute the reason for this observation.
10. How to reduce the glass transition temperature further, authors can state the few techniques through which the reduced glass transition temperature is achievable? In commercial cases, the soft polymeric systems exerts even lower glass transition temperature. How it is important for achieving the good optical properties?
11. I suggest authors to re-evaluate the stress-strain characteristics.
12. Supplementary Figure 44 graph contrasts the merits of the as-obtained materials. The graphical representation failed to reveal the references in the graphical picture. Add reference in to graph.
13. Elasticity of the as-obtained phosphorescent composite should be mentioned. Recovery aspects and residual strain characteristics can be studied to differentiate the various systems.
14. Upon stretching, lifetime values tend to degrade, why such trends are observed. Authors can add some light to it.
15. PABE-d materials exerts underwater phosphorescence. Why this is important, and how the lifetime values are recovered upon annealing at 70°C under vacuum. Why specifically vacuum conditions are imposed, what would be the effects of various humidity or temperatures?

16. How the hydrogen bonds contributes to the phosphorescent characteristics? How do authors gauge the extent of hydrogen bond formation within the matrix? Authors can propose the mechanistic scheme for the better understanding.
17. Absence of clear schematic and mechanism seriously degrades the quality of the presented work. Authors should consider revising the entire article to address the grammar and typographical errors.
18. I suggest authors to cite some of the recent related literatures. Authors can cite recent references, which can deliver and connect the audience with the current study. Authors can refer the following recently published articles and strengthen introduction and the reference list,

Nature Materials (2023). <https://doi.org/10.1038/s41563-023-01703-0>

Advanced Science, 2021, 8 (21), 2102275. <https://doi.org/10.1002/advs.202102275>

Journal of the American Chemical Society, 2021, 143, 34, 13675–13685.
<https://doi.org/10.1021/jacs.1c05213>

Advanced Materials, 2023, 35, 2303267. <https://doi.org/10.1002/adma.202303267>

ACS Nano 2022, 16, 10, 16201–16210. <https://doi.org/10.1021/acsnano.2c05056>

Chemical Engineering Journal, 418, 2021, 129421. <https://doi.org/10.1016/j.cej.2021.129421>

ACS Applied Materials & Interfaces, 2023, 15, 28, 33784–33796.
<https://doi.org/10.1021/acsami.3c03794>

ACS Applied Materials & Interfaces, 2020, 12, 12, 14408–14415.
<https://doi.org/10.1021/acsami.9b23291>

All the best for the revision. After receiving the convincing point-to-point responses, I will recommend it for the publication.

Point-by-Point Response to Reviewers

Reviewer #1:

This work reported a kind of block copolymer based on multiphase engineering. The as-prepared polymer showed both rigidity and flexibility, and a room temperature phosphorescent material with long-lived emission and high stretchability was acquired. In addition, the authors demonstrated the application of this material in multi-level volumetric data encryption and stretchable afterglow display. This is an interesting work with sufficient data. I recommend its publication after revisions, and there are some confusions need to be solved.

Response: We sincerely appreciate the reviewer for the positive comments on our manuscript and careful review to improve the quality of our paper.

Comment #1: How many times are the measurements for the stress-strain curves? I recommend that at least three times parallel measurements are implemented for the curves in Fig. 2, and the error bar should be provided. Moreover, what is the detailed measurement information for the mechanical measurements?

Response: We are sincerely grateful for the reviewer's suggestion. We have carried out five times of parallel mechanical measurements for each sample. As suggested by the reviewer, the error bars have been added in **Figure 2e** in the revised Manuscript, which represents the standard deviation calculated by three times data sets (the highest and lowest data were discarded). The mechanical properties of polymers were tested by a universal testing machine (Instron 3344) with rectangular specimens of polymer films with dimensions of 30 mm × 10 mm × 0.5 mm at a tensile rate of 5 mm min⁻¹ at room temperature. The detailed measurement information has been added in the revised Supporting Information.

Supporting Figure 1. Toughness and Young's modulus variation of different block polymers under ambient conditions. Error bars represent the standard deviation calculated by three times data sets.

Comment #2: The authors have mentioned the good water resistance for the copolymer. What are the performances for these copolymers in water? Would the water affect the hydrogen bonds and thus the RTP performances?

Response: We appreciate the comment from the reviewer. Taking PABE-d film as an example, we tested the RTP performance in water solution. As shown in **Supplementary Figure 58**, as time increased from 0 to 120 min, the phosphorescence intensity of PABE-d film showed a decrease, with lifetime decreasing from 221.0 to 171.0 ms. The RTP lifetime of the soaked film can be recovered to 220.6 ms after being heated under vacuum at 70 °C for 2 h. These data demonstrated that the water could affect the hydrogen bonds and thus weaken the RTP performances to some extent, but the ultralong RTP can still be observed within 120 min. It suggested that the water could affect hydrogen bonds between PAA chains but would not destroy the hydrogen-bond network absolutely. This benefitted from the hydrophobic PBMA segment, which could resist and delay the entry of water molecules into the PAA segment and thus protected phosphors for improving the water resistance of these copolymers. After heating, the immersed water molecules were removed from the polymer matrix, and the rigid hydrogen-bond networks between polymer chains and chromophores were restored. Thus, the ultralong RTP lifetime of PABE-d can be recovered to the initial state. The related data of PABE-d film in water was added to the revised Supporting Information.

Supplementary Figure 58. (a) Phosphorescence spectra and (b) lifetime profiles of the emission band at 548 nm of PABE-d film after being soaked in water with different times ranging from 0 to 120 min. (c) Lifetime profile of the soaked film after being heated under vacuum at 70 °C for 2 h.

Comment #3: Why selected the naphthalimide derivative as the chromophores in this work?

Response: We appreciate the comment from the reviewer. Naphthalimide and its derivatives possess multiple carbonyl groups and hetero atoms (like N, O), which could effectively facilitate the population of triplet states through an efficient intersystem crossing process. Moreover, these molecules showed rigid planar molecular skeletons that can restrict molecular rotations/vibrations for suppressed non-radiative decay of triplet states (*Angew. Chem. Int. Ed.* **2016**, 55, 9872; *J. Am. Chem. Soc.* **2018**, 140, 1916; *Nat. Commun.* **2022**, 13, 4890). Therefore, we select naphthalimide derivatives as phosphorescence chromophores in our manuscript. To

give a better understanding to the audience, we added relevant illustrations on molecule design in the introduction part in the revised Manuscript.

Comment #4: The authors have demonstrated the function of hydrogen bonds in enhancing the RTP emissions. Are there any referenced experiments to validate this demonstration?

Response: We are grateful for the constructive question from the reviewer. In our manuscript, we carried out some referenced experiments to reveal the function of hydrogen bonds in enhancing RTP emissions (**Supplementary Figure 74**). To further demonstrate the function of hydrogen bonds, we performed control experiments by selecting polymer PBM as a model (**Supplementary Figure 72**). Before hydrolyzing, almost no RTP phenomenon can be observed from PBM under ambient conditions. This result demonstrates that the PBM matrix cannot effectively protect triplet excitons due to the lack of carboxyl groups to form strong hydrogen bonds for suppressing non-radiative decay. In contrast, after hydrolyzing, ultralong phosphorescence can be easily observed from polymer HPAA under ambient conditions (**Supplementary Figure 43**). Because after hydrolyzing, there formed numerous carboxyl groups that can generate strong hydrogen bonds between polymer chains and phosphors. Such a hydrogen bonding network can provide a rigid microenvironment to restrict molecular motions of phosphors for effectively suppressing the non-radiative transition, thus generating ultralong RTP. The relevant experiments have been included in the revised Supporting Information.

Supplementary Figure 74. Photoluminescence (black lines) and phosphorescence (red lines) spectra of control polymers (a) PBBE-a and (b) PBMA films under ambient conditions. Lifetime profiles of the emission band at 377 nm for (c) PBBE-a and (d) PBMA films.

Supplementary Figure 72. Steady-state photoluminescence (black line) and phosphorescence (red line) spectra of polymer PBM under ambient conditions.

Supplementary Figure 43. (a) Normalized steady-state photoluminescence (PL) and phosphorescence (Phos.) spectra of HPAA film under 340 nm excitation. (b) Lifetime decay curve of emission band at 548 nm. (c) Lifetime profile of emission band at around 394 nm.

Comment #5: To prepare the films, the copolymers were hot-pressed under 120 °C. Would this process affect the molecular arrangement or RTP performances of the materials?

Response: We thank the reviewer's comment. In this work, the hot-press process was utilized for obtaining large area and uniform polymer films for subsequent mechanical properties tests. To eliminate the confusion of reviewers, block copolymer films were also prepared by a rotary evaporation method. The photophysical properties of these polymer films were also investigated by a set of further experiments. As shown in **Supporting Figures 2-3** and **Supporting Table I**, these copolymer films displayed similar PL and RTP emission wavelengths, lifetimes, and quantum efficiencies as the hot-expressed films. From these data, the hot-pressed process had little influence on the RTP performances of the copolymers. As the reviewer stated, the hot-press process might affect the molecular arrangement of polymer, but it has a slight effect on the RTP performance in this research.

Supporting Figure 2. Normalized steady-state PL (blue lines) and phosphorescence (red lines) spectra of PAMA, PAME, PAPE, and PABE copolymer films prepared by rotary evaporation.

Supporting Figure 3. Lifetime profiles of the emission bands at 548 nm for PAMA, PAME, PAPE, and PABE copolymer films prepared by rotary evaporation.

Supporting Table 1. Photoluminescence efficiencies of the as-prepared PAMA, PAME, PAPE, and PABE films under ambient conditions.

Polymers	PAMA	PAME	PAPE	PABE
$\Phi_{\text{Fluor.}}$ [%]	11.35	15.04	11.19	15.77
$\Phi_{\text{Phos.}}$ [%]	1.75	1.86	1.51	2.33

Reviewer #2:

This manuscript introduced stretchable block copolymers with ultralong room temperature phosphorescence. It achieved a good stretchability of 712% and an ultralong phosphorescence lifetime of up to 981.11 ms. Although the author describes the difficulty of obtaining long afterglow and stretchable properties, there are still some relevant reports. Unfortunately, the RTP quantum yield here is not as good as these reports due to the phosphors used in this manuscript are very common. This manuscript does not demonstrate the importance of excellent stretchability in applications. In addition, the author spends a lot of time explaining the mechanism of stretchable phosphorescent copolymers. They said it can efficiently address the dilemma between the “rigidity” and “flexibility” in the same polymer. But the good stretchability requires sacrificing RTP QYs. The QYs were significantly decreased from 2.2% (PABE) to 0.97%(PABE-d). A similar work using only PAA can reach a phosphorescence quantum yields of 4.0% (Ref: *Journal of Luminescence* 263 (2023) 119978). And many stretchable RTP polymer have been reported previously (DOI: 10.1021/jacs.2c13264, 10.1002/adom.202102449). I do not recommend this manuscript accepted by such a high-level journal as *Nature Communication* because this work lacks importance and innovation.

Response: We truly understand this reviewer’s concern. As the reviewer mentioned, the RTP quantum yield or lifetime of a copolymer is significantly influenced by the choice of chromophore. In this research, we are focusing on developing polymers that exhibit long-lived RTP and simultaneous good stretchability. For this purpose, we selected chromophores with long phosphorescent lifetimes, such as MBe, MBi, PNa, DBI, and MPy as initiators. Generally, balancing the phosphorescence lifetimes and quantum yields of chromophores is a long-term challenge in this field (*Chem* 2016, 1, 592; *Nat. Rev. Mater.* 2020, 5, 869; *Ann. Phys. (Berlin)* 2019, 531, 1800482). To date, most reported long-lived RTP materials always exhibit lower phosphorescence quantum yield due to the limitation of excited state dynamics (Equitation 1-3).

$$\Phi_p = \Phi_{isc} k_p \tau_p \quad (1)$$

$$\Phi_{isc} = k_{isc} / (k_f + k_{ic} + k_{isc}) \quad (2)$$

$$\tau_p = 1 / (k_p + k_{nr}) \quad (3)$$

The stretchable polymers with high quantum yield can also be achieved using our universal strategy of multiphase engineering through tailoring or replacing the chromophores. However, this will sacrifice the RTP lifetime of the polymer. To verify this, we further designed and synthesized block copolymers PABr and PPBr utilizing Br-modified BMP and BEP initiators to improve the phosphorescence quantum yield (**Supplementary Figures 30-33** and **39**). As shown in **Supplementary Figure 89** and **Supplementary Table 16**, after introducing the Br atom on the initiators, the phosphorescence proportion in PL spectra showed a dramatic increase up to 91% and 96% for PABr and PPBr films, respectively. Benefitting from the heavy atom effect of the Br atom for enhanced spin-orbit coupling (SOC) and thus promoted intersystem crossing (ISC) process, the RTP quantum yields of PABr and PPBr films increased to 35.90% and 13.85%, maintaining a good stretchability of 653.8% and 633.1%, respectively. These data indicated that our proposed multiphase design

principle can also prepare stretchable copolymer films with high RTP yield, demonstrating the generality of our strategy.

In this work, combining the color-tunable ultralong RTP and high stretchability of block copolymers, **we demonstrated their potential in information storage and stretchable afterglow display**. Benefitting from the excellent stretchability of copolymers, the flexible display array can be utilized even under large mechanical deformation including bending and stretching. These materials will lay the foundations for expanding potential applications of RTP materials in stretchable photoelectronics, such as stretchable sensors and detectors, on-skin displays, and so forth.

Although some stretchable RTP materials have been reported, most of them are realized by physically doping phosphors into a flexible polymer matrix. **Furthermore, the stretchability of these doping materials is typically limited, and their mechanical performance is not adjustable**. Additionally, we have carefully read the references presented by the reviewer. For **Ref1** (Journal of Luminescence 263 (2023) 119978), phosphorescence quantum yields of NI/PVA and NI/PAA doped films were 4.2% and 4.0%, higher than that of PABEs copolymers in our work. However, the phosphorescence lifetimes of NI@PVA and NI@PAA were only 147.7 ms and 192.1 ms, respectively, lower than that of PABE film (231.36 ms) in our work. The high phosphorescence quantum yields in Ref1 require sacrificing RTP lifetimes, as stated above. It should be noted that **NI@PVA and NI@PAA films mentioned in Ref1 lack good stretchability** due to the high rigidity of PVA and PAA matrices. This is a common issue observed in most RTP homopolymers created by conventional radical polymerization or physical doping methods. That is also the reason why **we proposed multiphase engineering to design a series of block copolymers to address this dilemma between the “rigidity” and “flexibility” in the same polymer**. Moreover, quantum yield is not the only parameter to evaluate phosphorescence performance, which also contains emission lifetime, emission color, etc. As mentioned above, based on our multiphase design principle, the RTP performance of the copolymer can be facilely manipulated by choosing different initiators, whether high RTP quantum yields, ultralong RTP lifetimes, or tunable RTP colors. Block copolymers in our work showed high stretchability and long phosphorescence lifetime with tunable emission colors, which indeed addressed the dilemma between the “rigidity” and “flexibility” in a polymer system.

For **Ref2** (DOI: 10.1021/jacs.2c13264), the authors reported CE-PAAm composite hydrogels with clusterization-induced phosphorescence from carbonyl groups. There exist many differences between Ref2 and our work:

(1) **RTP performance**. The phosphorescence lifetime of CE-PAAm gel was 342 ms, while the phosphorescence quantum yield was **only 0.4%**. Only a small portion of the polymeric matrix is geometrically confined by the in situ-formed crystals, while a large fraction of carbonyl groups do not form clusters, which absorb excitation light yet do not emit phosphorescence and result in a very low RTP quantum yield. Meanwhile, the phosphorescence center of the composite hydrogel originated from carbonyl clusters formed and confined by the squeezing-out effect of in situ crystallization of CE dopants. This determines that the RTP performance regulation in

these gels is very limited. Although CE-PAAm gel displayed excitation-dependent phosphorescence owing to the carbonyl clusters, the maximum emission only changed from 450 to 510 nm. In our work, the RTP of copolymers originated from the isolate molecule phosphorescence of the initiator in the polymer matrix. The RTP performance can be well-tuned by tailoring structures of initiators, wherein the RTP lifetimes ranged from 220 ms to 981 ms in different copolymers, and **RTP colors can be tuned from 420 nm to 600 nm**, covering almost the entire visible range. As discussed above, by introducing Br-modified initiators, high RTP quantum yields can also be realized. Therefore, the RTP copolymer prepared by multiphase engineering in our work endows **a high tunability on the RTP performance**.

(2) **Mechanical property**. Although CE-PAAm gel showed good stretchability, the tunability of mechanical performance was not discussed because these hydrogels were prepared by conventional radical polymerization, which was not controllable as the ATRP method presented in our work. The “living” character of ATRP allows for precise control of molecular architecture and molecular weight, giving rise to further tuning the mechanical performance of block copolymers. In our work, **block copolymers not only displayed good stretchability, but their mechanical properties were also well-tuned by changing molecular architecture and molecular weight of the soft block**. For DBI-based copolymers, the intrinsic stretchability can be tuned from 10.5% to 689.5%, Young’s modulus can be tuned from 676.6 MPa to 15.4 MPa, the strength-at-break can be tuned from 48.7 MPa to 3.4 MPa, and the toughness can be tuned from 33.8 to 3.2 MJ m⁻³. We have systematically studied the relationship between copolymer structures and RTP as well as mechanical performance, which will **give some guidance to choosing phosphors and soft block structures for designing high-performance stretchable phosphorescent polymers in other systems**.

(3) **Mechanism of stretchable phosphorescent materials**. The mechanism presented in Ref2 was unclear. As the authors described in their article, “Considering the complexity of carbonyl clusters in structure, conformation, and energy-level distribution, it is still a great challenge to precisely reveal the phosphorescence mechanism.” Therefore, the authors provided a plausible explanation based on the FTIR data. In our work, to reveal the relationship between nanostructure morphology and macroscopic polymer properties, we carried out various characterizations, including SAXS, TEM, WAXS, confocal fluorescence microscopy, XRD, and a series of control experiments. Besides, we performed large-scale coarse-grained molecular dynamics simulations to further reveal the self-assembly behavior of these block copolymers. Combining experimental and theoretical calculations, we found that the high stretchability and simultaneous ultralong phosphorescence were ascribed to the formation of soft-hard microphase-separated structures in these copolymers. **This work will provide a clear and fundamental understanding of the nanostructures and material properties for designing high-performance stretchable RTP materials**.

For **Ref3** (10.1002/adom.202102449), the authors reported strain-responsive RTP and investigated material performance through the phosphorescence lifetime and

image analysis (PLIA) technique. There exist many differences between Ref3 and our work:

(1) **Materials preparation and performance.** PA6/TPAB in Ref3 was prepared by melt blending approach upon doping chromophores into a polymer matrix. Phase separation is a common challenge in this method. The melting temperature and doping concentration of chromophores limited the universality of this method to some extent. Meanwhile, polymer PA6 was purchased from commercial sources. Thus, the structure of PA6/TPAB was fixed, and the mechanical performance was not as tunable as our work. The elongation of PA6/TPAB was only 250%, much lower than our work with a good stretchability of 712%. In our work, the copolymers were prepared by ATRP, which is applicable to a large range of monomers, initiators, and dispersed media. The diversity of initiator and monomer gives rise to a high tunability on the RTP and mechanical performance of the block copolymers in our work, as well as the good universality of our method.

(2) **Mechanism.** Ref3 reported strain-responsive RTP with more emphasis on describing this dynamic phenomenon, and the related process was investigated using the PLIA technique, while deep mechanism explanation and related proof were scarce in the paper. In our work, we carried out a lot of characterizations to explain and demonstrate the mechanism of stretchable phosphorescent copolymers, deeply revealing the relationship between nanostructures and material properties.

(3) **Application.** The authors mentioned that strain-responsive RTP was useful for deformation sensing and early damage reporting of engineering materials, but they didn't show any detailed application demonstration in the paper. In our work, combining the color-tunable ultralong RTP and high stretchability of block copolymers, **we first demonstrated their potential in multi-level volumetric data encryption, information storage, and stretchable multicolor afterglow display.** The stretchable copolymer films can be utilized for large-area full-color afterglow display (5×5-pixel array), even under large mechanical deformation including bending and stretching. These applications benefitted from the excellent stretchability of copolymers and had never been demonstrated in previously reported RTP materials. These results could lay a foundation for extending the potential of phosphorescence polymers to new domains.

In conclusion, we **first present a universal strategy to realize high-performance stretchable phosphorescent materials by multiphase engineering.** The mechanical and phosphorescence performance of resulting block copolymers can be well-tuned by varying soft blocks and initiators. The detailed mechanism exploration provides a fundamental understanding of the nanostructures and material properties for designing high-performance stretchable materials. This **pioneering multiphase engineering by ATRP method in the organic RTP field** will inspire researchers to explore more intrinsically stretchable phosphorescent materials and other multifunctional flexible luminescent materials by manipulating various blocks or phosphors, which would present a significant technological advance in expanding the scope of luminescence materials for organic

optoelectronics. We hope our candid clarification will be able to convince the reviewer of the importance and innovation of our work.

Supplementary Figure 30. Chemical structure of BMP initiator and the ¹H NMR spectrum in CDCl₃.

Supplementary Figure 31. ¹³C NMR spectrum of BMP initiator in CDCl₃.

Supplementary Figure 32. Chemical structure of BEP initiator and the ^1H NMR spectrum in CDCl_3 .

Supplementary Figure 33. ^{13}C NMR spectrum of BEP initiator in CDCl_3 .

Supplementary Figure 39. Chemical structures and GPC traces of block copolymers PABr and PPBr, respectively.

Supplementary Figure 89. (a) Normalized steady-state photoluminescence (dotted lines) and phosphorescence (solid lines) spectra for PABr and PPBr films under ambient conditions. Inserts show their RTP efficiencies. (b) Lifetime decay curves of emission bands at 490 and 585 nm for PABr and PPBr films, respectively. (c)

Stress-strain curves of PABr and PPBr films at room temperature.

Supplementary Table 16. Photoluminescence efficiencies of PABr and PPBr films under ambient conditions.

Polymers	$\lambda_{\text{Fluor.}}$ (nm)	$\Phi_{\text{Fluor.}}$ (%)	$\lambda_{\text{Phos.}}$ (nm)	$\Phi_{\text{Phos.}}$ (%)
PABr	328	1.50	490	35.90
PPBr	413	1.35	585	13.85

Comment #1. The authors observed the increased distance between adjacent emitters through confocal fluorescence microscopy. But the distance between adjacent emitters is at the nanoscale by TEM, which far exceeds the resolution of a fluorescence microscope. It cannot “matched the distribution of the PAA microphase in the TEM images”.

Response: We are grateful for the careful review from the reviewer. We apologize for our inaccurate description. Although their resolutions have significant differences, the variation tendency of the PAA phase distribution in TEM was in accordance with that of the emission areas in confocal fluorescence microscopy. To avoid misleading the reader, we revised the relevant illustrations: “As the content of the hard block decreased, the emission areas became sparse, which **accorded with the variation tendency** of the PAA microphase distribution in the TEM images”. The corrected sentence was highlighted in the revised Manuscript.

Comment #2. The final polymer was precipitated in MeOH/H₂O mixtures. Why the hydrophilic PAA is located inside the self-assembled structure, while hydrophobic PBM is located outside?

Response: We thank the comments of the reviewer. As shown in **Supporting Figure 4**, the polymer PBBE precipitated in MeOH/H₂O mixtures is not the final product. To obtain the final amphiphilic block copolymer PABE, polymer PBBE needs to be further dissolved in dichloromethane (DCM) and added with trifluoroacetic acid (TFA). This mixture needs to be stirred at room temperature for 24 hours to convert the *tert*-butyl ester into a carboxyl group. Once the hydrolyzation process in the DCM solution is completed, the amphiphilic block copolymer PABE will self-assemble into two-phase nanostructures. The PBMA block will form a continuous phase (outside) due to being in a good solvent DCM, while the PAA block will form a dispersed phase (inside) in a poor solvent DCM. The detailed illustration was presented in **Supplementary Figure 2** and “**Synthesis of block copolymers**” in Supporting Information.

Supporting Figure 4. Synthetic routes of amphiphilic block copolymer PABE.

Comment #3. Does TGA curves indicate solvent residue in the polymer?

Response: We sincerely appreciate the careful review of the reviewer. To eliminate the effects of residual solvents, we further dried samples under a vacuum at 70 °C for 48 h. After re-evaluating the thermal analysis data, TGA curves of other polymers are similar to the previous results. The TGA data of polymers HPAA and PBMA showed a little variation, which might be ascribed to the residual solvent in polymers. The corrected data were added in our revised Supporting Information.

Supplementary Figure 56. TGA curves of copolymers PABEs, PBIEs, PNPys, and control polymers HPAA and PBMA, respectively.

Comment #4. Does the phosphor will be brominated by the bromine radical, which is generated by the ATRP process?

Response: We thank the comments of the reviewer. The phosphor will not be brominated by the bromine radical during the ATRP process. According to the reaction mechanism of ATRP, this process is controlled by an equilibrium between propagating radicals (R[•]/P_n[•]) and dormant species, predominately in the form of

initiating alkyl halides or macromolecular species (RX or P_nX). The related process was presented in **Supporting Scheme 1** (*Macromolecules* **2012**, *45*, 4015; *Nat. Chem.* **2009**, *1*, 276). The dormant species periodically react with the rate constant of activation (k_{act}) with transition metal complexes in their lower oxidation state, Mt^m/L, acting as activators (Mt^m represents the transition metal species in oxidation state m and L is a ligand; the charges of ionic species and counterions are omitted here) to reversibly form growing radicals (P_n^{*}), and deactivators—higher-oxidation-state metal complexes with coordinated halide ligands X-Mt^{m+1}/L. The radically transferable halogen atom was generally attached to the terminal of the obtained polymer. To demonstrate our statement, we further synthesized polymer PBM with a low molecular weight of 4670 Da following the same ATRP method (**Supporting Figure 5**). The chemical structure was characterized by ¹H NMR spectra (**Supporting Figure 6**). Compared with the DBI initiator, PBM polymer showed a similar chemical shift and integral area of naphthalimide moiety (in the range of 9.0–7.0 ppm). These results indicated that the phosphor was not brominated by the bromine radical during the ATRP process.

Supporting Scheme 1. The reaction scheme for ATRP. P_n represents polymer (with degree of polymerization n), Mt^m is the metal species in oxidation state m, L is a ligand and X is a halogen atom. M represents monomer. The kinetic parameters k_{act} , k_{deact} , k_p and k_t represent the rate constants of activation, deactivation, propagation, and termination, respectively.

Supporting Figure 5. GPC trace of polymer PBM used for NMR test.

Supporting Figure 6. (a) ¹H NMR spectra of DBI initiator and PBM polymer in CDCl₃. (b) Enlarged ¹H NMR spectra of DBI initiator and PBM polymer in the range of 9.0-7.0 ppm.

Comment #5. How does the binary initiator system work? How about the simple mixture of afterglow?

Response: According to the reaction mechanism of ATRP, in the binary initiator system, the polymer chains are initiated separately by two initiators DBI and MBe until termination reactions occur and form the hybrid system of polymers PABE and PAPh (**Supporting Scheme 2**). Furthermore, GPC analysis of the binary initiator

system showed that it does not affect the "living" characteristic of the ATRP method. This means that block copolymers can maintain targeted composition, controlled molecular architecture, predetermined molecular weight, and narrow PDI like the single initiator system PABEs. By tailoring the initiator components and the feed ratio of different initiators, the afterglow of copolymers can be easily adjusted. More importantly, the binary initiator system exhibited much better uniformity for the RTP emission. In addition, as the reviewer suggested, we have also mixed copolymers PABE and PAPh using physical doping. However, obtaining a uniform film through physical doping is challenging due to the poor solubility of the amphiphatic block copolymers PABE and PAPh, particularly for polymers with higher molecular weight. After mixing the two copolymers, although the resulting polymer also showed a color-tunable afterglow due to containing two different phosphors, the luminescence is nonuniform across the film (**Supporting Figure 7**). Therefore, in our manuscript, we adopted the binary initiator method to obtain the color-tunable RTP materials.

Supporting Scheme 2. The composition of the binary initiator system. Note: the polymer chains that participated in termination reactions were ignored.

Supporting Figure 7. Photographs of PABE-PAPh co-doping film ($m_{\text{PABE}}: m_{\text{PAPh}} = 1:30$) taken under daylight and after turning off the 254 and 365 nm UV lamps.

Comment #6. What's the ratio of the two phosphors in the binary initiator system? It's unlikely to be the feed ratio.

Response: We sincerely appreciate the critical question from the reviewer. To eliminate the puzzle of the reviewer, with the PDN as a model (**Supplementary Figure 4**, the feed ratio is DBI: MBe=1:30), we measured the UV-vis absorption spectra of relevant initiators (DBI and MBe) and polymer PDN in dichloromethane (DCM) solution to determine the ratio of two phosphors in the binary initiator

system. Specifically, the absorption spectra of initiators DBI and MBe with concentrations ranging from 0.01 to 0.08 mg/mL and the polymer PDN of 1.0 mg/mL were collected under ambient conditions (**Supporting Figure 8a-c**).

Based on **Supporting Figure 8a**, the standard job's plots of absorbance of DBI at 334 nm and 250 nm versus concentrations (M) can be well fitted as (**Supporting Figure 8d-e**):

$$y_{\text{DBI}(334 \text{ nm})} = 35.87619x - 0.07043 \quad (1)$$

and

$$y_{\text{DBI}(250 \text{ nm})} = 6.4619x - 0.00654 \quad (2)$$

in which y and x refer to the absorbance and concentration, respectively.

From **Supporting Figure 8c**, the absorbance (y) value of 1 mg/mL polymer PDN at 334 nm is 0.015. And the absorption at 334 nm was contributed by DBI as MBe showed no contribution at 334 nm. Therefore, the phosphor DBI concentration in PDN solution could be calculated to be $M_{\text{DBI}} = 2.38 \times 10^{-3}$ mg/mL according to Equation (1), and the corresponding molar concentration is $C_{\text{DBI}} = 6.12 \times 10^{-6}$ M.

Given the determined DBI concentration in polymer PDN, the absorbance of PDN at 250 nm contributed by DBI could be obtained based on Equation (2). And the specific value is $y_{\text{DBI}(250 \text{ nm})} = 0.00885$. While the experimentally collected absorbance of PDN at 250 nm is $y_{\text{PDN}(250 \text{ nm})} = 2.055$, hence, the absorption of PDN contributed by MBe at 250 nm is determined as $y_{\text{MBe}(250 \text{ nm})} = 2.0462$ based on Equation 3:

$$y_{\text{PDN}(250 \text{ nm})} = y_{\text{DBI}(250 \text{ nm})} + y_{\text{MBe}(250 \text{ nm})} \quad (3)$$

From **Supporting Figure 8b**, the standard job's plot of absorbance of MBe versus concentration can be fitted as (**Supporting Figure 8f**):

$$y_{\text{MBe}(250 \text{ nm})} = 30.32738x + 0.66889 \quad (4)$$

Therefore, the concentration of MBe in PDN is $M_{\text{MBe}} = 4.54 \times 10^{-2}$ mg/mL and the corresponding molar concentration is $C_{\text{MBe}} = 1.514 \times 10^{-4}$ M.

Hence, the molar ratio of DBI: MBe is $C_{\text{DBI}} / C_{\text{MBe}} = 1/25$, which is very close to the feed ratio of PDN (DBI/MBe=1/30).

Supporting Figure 8. UV-vis absorption spectra of (a) DBI and (b) MBe initiators in DCM solution at room temperature (0.01-0.08 mg/mL). (c) UV-vis absorption spectra of PDN polymer in DCM solution at room temperature (1.0 mg/mL). Absorbance-concentration fitted plots of DBI initiator at (d) 334 nm and (e) 250 nm, and that of (e) MBe initiator at 250 nm.

Comment #7. The impurity shows great influence on the phosphorescence, especially afterglow (10.1038/s41563-020-0797-2). The origin of the afterglow should be further investigated. A strict purification was suggested for the phosphor-based initiators. Then, the phosphor-based initiators are suggested to dope into polyacrylic acid physically to testify whether the afterglow is generated by the initiator.

Response: We truly understand this reviewer's concern. As suggested by the reviewer, all the phosphor-based initiators were strictly purified through column chromatography and recrystallization. Their chemical structure and purity were further thoroughly characterized and confirmed by ^1H NMR, ^{13}C NMR, high-performance liquid chromatography (HPLC), mass spectra, and elemental analysis.

Additionally, to get deep insights into the origin of afterglow, as the reviewer suggested, we prepared a series of hybrid polymer films by doping the phosphor-based initiators into the polyacrylic acid (PAA) matrix with a doped concentration of 0.5%. The photophysical properties of these films were investigated by the photoluminescence spectra and lifetime decay profiles. As depicted in **Supplementary Figure 66**, the hybrid systems showed RTP with peaks at 416, 479, 512, 548, and 602 nm for MBe-PAA, MBe-PAA, PNa-PAA, DBI-PAA, and MPy-PAA, respectively. Moreover, the phosphorescence bands all exhibited long lifetimes ranging from 219.32 to 1172.56 ms (**Supplementary Figure 67**). These results are well consistent with the corresponding block copolymers. We also collected the phosphorescence spectra of these initiators in 2-methyl-tetrahydrofuran (2-mTHF) solution (10^{-5} M) at 77 K (**Supplementary Figure 68**). The phosphorescence spectra of these initiators are consistent with block copolymers and PAA-doped

hybrid systems, demonstrating that the phosphorescence of copolymers originated from the isolate molecule phosphorescence of initiator chromophores in the PAA matrix. These data eliminated the potential influence of impurities on the luminescence properties. All the supplementary data were added in the revised Manuscript and Supporting Information.

Supplementary Figure 34. High-performance liquid chromatogram spectra of MBe, MBI, PNa, DBI, and MPy initiators in acetonitrile solution.

Supplementary Figure 66. Photoluminescence (blue lines) and phosphorescence (red lines) spectra of different hybrid systems (a) MBe-PAA, (b) MBI-PAA, (c) PNa-PAA, (d) DBI-PAA, and (e) MPy-PAA under ambient conditions.

Supplementary Figure 67. Phosphorescence decay profiles of different hybrid systems (a) MBe-PAA, (b) MBi-PAA, (c) PNa-PAA, (d) DBI-PAA, and (e) MPy-PAA under ambient conditions.

Supplementary Figure 68. UV-vis absorption (black lines) and photoluminescence (green lines) spectra of initiators in dilute DCM solution (10^{-5} M) under ambient conditions, and phosphorescence (red lines) spectra of initiators in dilute 2-methyltetrahydrofuran solution (10^{-5} M) at 77 K.

Reviewer #3:

Authors have successfully demonstrated the fabrication of stretchable block copolymer based phosphorescent materials. Importantly, the as-proposed synthetic methodology provides convenient optimization of the as-proposed block system with tunable properties. The research carried out seems to have significant impact and the proposed methodology is convincing. Although, I request authors to address the following comments, thereby suggesting the major revision for the present stage. I suggest authors to address the following comments.

Response: We sincerely appreciate the reviewer for the careful review to improve the quality of our paper.

Comment #1. Authors have emphasized the significance of framing the stretchable room temperature phosphorescent materials, but failed to stress the choice of selection of chromophore units. How do the as-presented chromophore materials fit in the desired synthetic methodology?

Response: We sincerely appreciate the critical question from the reviewer. For clarity, the photophysical process of organic room temperature phosphorescence was illustrated in **Supporting Figure 9** (*Appl. Phys. Rev.* **2023**, *10*, 021313). Upon photoexcitation, organic molecules can absorb the photons and generate singlet excitons. Then, the singlet excitons return to the ground state, releasing energy in the form of radiative decay, which is known as fluorescence. Notably, the singlet excitons can also transfer to triplet excited states via an intersystem crossing (ISC) process, populating triplet excitons. These triplet excitons then return to the ground state through radiative decay would generate phosphorescence, accompanied by a nonradiative decay process.

However, organic molecules usually possessed very weak spin-orbit coupling (SOC) between singlet and triplet states, resulting in the inefficient population of triplet excitons. Meanwhile, the triplet excitons are easily quenched by molecular rotation/vibration, oxygen, moisture, and other quenchers. Therefore, promoting the ISC process to produce triplet excitons and suppressing the non-radiative decay of triplet excitons are two prerequisites. Following this, recently, several molecular design rules were proposed, such as introducing heavy-atom (like Cl, Br, I), carbonyl group, and hetero-atoms (like N, O, S) into molecule skeletons to promote the ISC process for producing triplet excitons. On the other hand, suppressing the non-radiative decay of triplet excitons by constructing a rigid microenvironment, including crystal engineering, synthesizing rigid polymer matrix, building rigid MOF or COF, supramolecular assembly, and so on.

Following the abovementioned principles, we have chosen a series of chromophore units that contain hydroxy, carbonyl, and amide. These units containing hetero atoms (N and O) with lone-pair electrons can facilitate the ISC process to effectively populate triplet excitons. Moreover, these chromophores possess relatively rigid and planar molecular skeletons that are favorable for reducing molecule motions for suppressing non-radiative decay. Meanwhile, the hydroxy on the molecular skeleton can act as a reaction site with 2-bromoisobutyryl bromide,

enabling the chromophore unit to convert to chromophore initiator for the following ATRP process (**Supplementary Figure I**). The related illustration of molecule design was added in the revised Manuscript.

[REDACTED]

Supporting Figure 9. (a) Modified Jablonski diagram for the photophysical process of fluorescence and phosphorescence. (b) Principles to generate phosphorescence in organic materials by promoting the ISC process and suppressing non-radiative decay.

Supplementary Figure I. Synthetic routes of initiators MBe, MBi, PNa, DBI, and MPy.

Comment #2. The discussion part with respect to synthesis and its corresponding characterization is limited. Authors should impart some considerable portion with respect to synthetic portions.

Response: We sincerely appreciate the reviewer's suggestion. The description of the synthetic portion was added and highlighted in the revised Manuscript:

Multiphase block copolymers were prepared by a two-step ATRP and subsequent hydrolyzation process (Supplementary Figs. 1-5). Firstly, we synthesized bromoisobutyrate-modified naphthalimide (DBI) as the ATRP initiator. Then, macroinitiator PBM was synthesized with DBI, *tert*-butyl acrylate, CuBr₂/TPMA (catalyst), and Cu(0) mixtures under nitrogen at room temperature, followed by next chain extension utilizing PBM, alkyl methacrylate monomer, CuCl₂/PMDETA (catalyst), and stannous octoate in a given ratio. After the reaction finished, the mixture was precipitated in methanol/H₂O to give the as-prepared block copolymer, which was further dissolved in dichloromethane/trifluoroacetic acid mixture and hydrolyzed at room temperature for 24 h to give the final amphiphilic block copolymer. Considering the alkyl side chain lengths will influence T_g and thus chain dynamics of corresponding copolymers, we selected methyl methacrylate (MMA), ethyl methacrylate (EMA), propyl methacrylate (PMA), and butyl methacrylate (BMA) as the second block monomers for preparing amphiphilic block copolymers PAMA, PAME, PAPE, and PABE, respectively. The chemical structures and purity of ATRP initiators were fully confirmed by nuclear magnetic resonance (¹H and ¹³C NMR), high-performance liquid chromatography-mass spectrometry (HPLC-MS), and elemental analyses (Supplementary Figs. 6-34). As demonstrated by Fourier transform infrared spectroscopy (FTIR) spectra, the hydroxy stretch at around 3100-3600 cm⁻¹ indicated that the *tert*-butyl ester groups were converted into carboxyl moieties after hydrolyzation, and the shift of carbonyl stretching resonance verified the formation of the C=O...H-O hydrogen bonds (Supplementary Fig. 35). The resulting polymer molecular weight and polydispersity (PDI) were determined by gel permeation chromatography (GPC) (Supplementary Figs. 36-39 and Supplementary Tables 1-3).

Comment #3. Why ATRP method is specifically chosen? What are the challenges in other methods? What are the specific limitations in ATRP synthesis?

Response: We thank the comments of the reviewer. In our manuscript, we designed and synthesized a series of block copolymers to form the microphase separation for solving the dilemma between the "rigidity" and "flexibility" in the same polymer. As we know, ATRP and reversible addition-fragmentation chain transfer (RAFT) are two well-established techniques to synthesize gradient copolymer with predictable molecular weight, relatively narrow dispersity, and high chain-end functionality under a mild reaction condition (*Nat. Rev. Chem.* **2021**, *5*, 859; *Nat. Chem.* **2009**, *1*, 276). In particular, ATRP is versatile, allowing for the use of a broad range of monomers and (macro)initiators, including (meth)acrylates, styrenes, acrylonitrile, and less reactive vinyl acetate and vinyl chloride. ATRP also employs many readily available alkyl halides as initiators, as well as commercially produced transition metal compounds and ligands. However, acidic monomers are not suitable for ATRP and require

protection or neutralization, which is a significant limitation. Moreover, ATRP requires the removal of a catalyst, a transition-metal complex with various ligands, from the final product via a complicated process.

On the other hand, RAFT has similar advantages to ATRP, but with some limitations. RAFT uses thioesters as mediating species, which are quite limited in availability due to their complex synthesis and high cost. Besides, the presence of thioester in the final polymer chain end can result in a colored product (yellow, brown, or red) that affects the optical properties of copolymers. In addition, compared with ATRP and RAFT, using other polymerization methods to obtain a well-defined copolymer with controllable and predetermined molecular weight and narrow PDI is challenging.

All things considered, we selected ATRP to synthesize the desired multifunctional block copolymers, as it provides a controllable and predetermined molecular weight and narrow PDI for achieving high-performance stretchable RTP materials.

Comment #4. The reason for selecting the hard phase PAA is vague. How do authors believe that PAA phase can prevent the triplet exciton quenching behavior?

Response: We apologize for our unclear statement. Generally, selecting a polymer matrix containing rich carboxyl groups, hydroxy groups, amido units, or other polar groups, such as polyacrylic acid (PAA), polyvinyl alcohol (PVA), polyacrylamide (PAM), polyacrylonitrile (PCN) and so on, is considered as a common approach to stabilize the triplet excitons of chromophores for generating phosphorescence. These groups can efficiently restrict molecular motions and suppress the non-radiative transition of triplet excitons by forming abundant hydrogen bonds between polymer chains and phosphors. Furthermore, these compact polymer chains can also protect the triplet excitons from being quenched by surrounding oxygen (*Nat. Commun.* 2020, 11, 944; *J. Am. Chem. Soc.* 2021, 143, 18527; *Nat. Commun.* 2022, 13, 3995; *J. Lumin.* 2023, 263, 119978). The relevant explanation for selecting PAA as the hard phase was added in the revised Manuscript.

To further demonstrate PAA phase can prevent triplet exciton quenching for RTP generation, we carried out a set of control experiments by selecting polymer PBM as a model. As shown in **Supplementary Figure 72**, before hydrolyzing, almost no RTP phenomenon can be observed from PBM under ambient conditions. This result demonstrates that the PBM matrix cannot effectively prevent the triplet exciton quenching behavior due to the lack of carboxyl groups to form strong hydrogen bonds for suppressing non-radiative decay. In contrast, after hydrolyzing, ultralong phosphorescence can be easily observed from polymer HPAA under ambient conditions (**Supplementary Figure 43**). Because after hydrolyzing, there formed numerous carboxyl groups that can generate strong hydrogen bonds between polymer chains and phosphors. Such a hydrogen bonding network can provide a rigid microenvironment to restrict molecular motions of phosphors for effectively suppressing the non-radiative transition, thus generating ultralong RTP. The related referenced experiments were added in the revised Supporting Information. Similarly, without a PAA hard block, control polymers PBBE-a and PBMA films exhibited almost no RTP under ambient conditions, demonstrating the importance of the PAA

hard phase in preventing the triplet exciton quenching behavior for obtaining ultralong RTP materials (**Supplementary Figure 74**).

Supplementary Figure 72. Steady-state photoluminescence (black line) and phosphorescence (red line) spectra of polymer PBM under ambient conditions.

Supplementary Figure 43. (a) Normalized steady-state photoluminescence (PL) and phosphorescence spectra of HPAA film under 340 nm UV light excitation. (b) Lifetime decay curve of emission band at 548 nm. (c) Lifetime profile of emission band at around 394 nm.

Supplementary Figure 74. Photoluminescence (black lines) and phosphorescence (red lines) spectra of control polymers (a) PBBE-a and (b) PBMA films under ambient conditions. Lifetime profiles of the emission band at 377 nm for (c) PBBE-a and (d) PBMA films.

Comment #5. Alkyl methacrylate soft phase is chosen, apart from this, there are many other soft monomeric units. Authors can mention various soft monomers and state the key reason for selecting the specific soft phase in the introduction part.

Response: We are sincerely grateful for the reviewer's suggestion. Generally, Soft monomeric units, including alkyl (meth)acrylates, carbosilane, siloxane, and ether chain, are widely used to decrease the T_g of polymer for improving the flexibility or stretchability (*J. Am. Chem. Soc.* **2022**, *144*, 4699). Particularly, alkyl methacrylate is a widely used monomer with high ATRP activity and can be commercially produced without additional synthesis. Meanwhile, in our manuscript, nonconjugated alkyl methacrylate monomer with high energy levels can prevent the excited state energy transfer from chromophore to polymer matrix for reducing phosphorescence quenching. Furthermore, alkyl methacrylate with the hydrophobic group helps to prevent chromophores from being quenched by surrounding moisture and improves the water resistance of block copolymers. As suggested by the reviewer, the related illustration of molecule design was added in the introduction part of the revised Manuscript.

Comment #6. Supplementary table I results reveals that upon increasing the chain length, the resultant polymer's molecular weight is increased. Interestingly, the PDI values experiences some irregular trends, why it is so?

Response: We sincerely appreciate the careful review of the reviewer. Despite ATRP is a controlling radical polymerization method, there are so many complicated factors that will strongly influence the reaction rate and dispersity of molecular weight (M_w/M_n), as shown in Equation 5 (*Macromolecules* 2012, *45*, 4015):

$$\frac{M_w}{M_n} = 1 + \frac{1}{DP_n} + \left(\frac{k_p[P_nX]}{k_{deact}[X-Cu^{II}/L]} \right) \left(\frac{2}{p} - 1 \right) \quad (5)$$

In the ideal case for fast initiation and no chain termination or chain transfer, several factors would affect the PDI of polymers prepared by ATRP, including the concentration of dormant species (initiating alkyl halides/macromolecular species, P_nX) and deactivator (transition metal complexes in higher oxidation state, $X-Cu^{II}$), the rate constants of propagation (k_p) and deactivation (k_{deact}), monomer conversion (p), degree of polymerization (DP_n), and reaction conditions (such as solvent, temperature, pressure, time, and stirring speed). Even slight variations in these parameters can influence the PDI values. The GPC results of five parallel polymerization experiments are presented in **Supporting Figure 10** and **Supporting Tables 2-5**. Despite our efforts to maintain the same experimental operation and reaction conditions, the PDI values showed irregular trends.

Supporting Figure 10. GPC traces of copolymers PBAEs, PBMEs, PBPEs, and PBBEs obtained from five parallel polymerization experiments.

Supporting Table 2. GPC characterizations of copolymer PBAEs obtained from five parallel polymerization experiments.

Sample	M_w	M_n	PDI
PBAE-1	71110	56510	1.25
PBAE-2	69950	57270	1.22
PBAE-3	72750	56660	1.28
PBAE-4	67840	54260	1.25
PBAE-5	74310	58520	1.27

Supporting Table 3. GPC characterizations of copolymer PBMEs obtained from five parallel polymerization experiments.

Sample	M_w	M_n	PDI
PBME-1	75520	59060	1.28
PBME-2	77080	59870	1.29
PBME-3	78600	60530	1.30
PBME-4	75970	60220	1.26
PBME-5	74540	58450	1.28

Supporting Table 4. GPC characterizations of copolymer PBPEs obtained from five parallel polymerization experiments.

Sample	M_w	M_n	PDI
PBPE-1	83060	61320	1.29
PBPE-2	73670	59340	1.24
PBPE-3	75200	60450	1.24
PBPE-4	77550	63000	1.23
PBPE-5	76380	62040	1.23

Supporting Table 5. GPC characterizations of copolymer PBBEs obtained from five parallel polymerization experiments.

Sample	M_w	M_n	PDI
PBBE-1	85230	68800	1.24
PBBE-2	83530	65220	1.28
PBBE-3	82030	65930	1.24
PBBE-4	88340	69700	1.27
PBBE-5	85400	68990	1.24

Comment #7. From Supplementary table 4, it is evident that PAPE shows the longer average lifetime value, but why PABE is considered for further studies?

Response: We are grateful for the careful review and question from the reviewer. From **Supplementary Table 4**, PAPE possessed the longest phosphorescence lifetime of 232.25 ms while PABE showed a shorter one of 231.36 ms. The difference between the two lifetimes is only 0.89 ms, which is nearly negligible for their long phosphorescence lifetimes over 200 ms. However, the stretchability of PABE of 188.0% is significantly higher than that of PAPE of 100.3%, displaying a significant increase of 87.4%. Considering the superior stretchability of PABE, we select PABE as a model polymer for further studies.

Supplementary Table 4. Luminescence lifetimes (τ) of PAMA, PAME, PAPE, and PABE films under ambient conditions.

Polymer	Wavelength (nm)	Fluorescence				Phosphorescence	
		τ_1 (ns)	A_1 (%)	τ_2 (ns)	A_2 (%)	τ (ms)	A (%)
PAMA	395	1.68	84.07	6.39	15.93	232.02	100%
	548						
PAME	395	1.62	82.23	5.45	17.77	223.86	100%
	548						
PAPE	395	1.53	78.15	6.10	21.85	232.25	100%
	548						
PABE	395	1.72	85.97	6.56	14.03	231.36	100%
	548						

Comment #8. The quantum yield of the as-obtained coblock systems are within the range of 1.2 to 2.2, Why the as-obtained quantum yields results are lower? Authors can attribute the reason for such lower quantum yield values. It is highly valued to know the control chromophore quantum yield values, so that audience can understand the significance of the as-obtained coblock system.

Response: We are grateful for the suggestion of the reviewer. The RTP quantum yield or lifetime of a copolymer is significantly influenced by the choice of chromophore. In this research, we are focusing on developing polymers that exhibit long-lived RTP and simultaneous good stretchability. For this purpose, we selected chromophores with long phosphorescent lifetimes, such as MBe, MBi, PNa, DBI, and MPy, as initiators. Generally, balancing the phosphorescence lifetimes and quantum

yields of chromophores is a long-term challenge in this field (*Chem* **2016**, *1*, 592; *Nat. Rev. Mater.* **2020**, *5*, 869; *Ann. Phys. (Berlin)* **2019**, *531*, 1800482). To date, most reported long-lived RTP materials always exhibit lower phosphorescence quantum yield. According to Equation 6-8,

$$\Phi_p = \Phi_{isc} k_p \tau_p \quad (6)$$

$$\Phi_{isc} = k_{isc} / (k_f + k_{ic} + k_{isc}) \quad (7)$$

$$\tau_p = 1 / (k_p + k_{nr}) \quad (8)$$

due to the weak spin-orbit coupling (SOC) and rapid rate of nonradiative decay in purely organic compounds, it is difficult to achieve phosphors with simultaneous efficiency and lifetime enhancement under ambient conditions. Stretchable polymer with high quantum yield can also be achieved using our universal strategy of multiphase engineering through tailoring or replacing the chromophores. However, this will sacrifice the RTP lifetime of the polymer. To verify this, we further designed and synthesized two block copolymers PABr and PBr with high quantum yields by utilizing Br-modified initiators BMP and BEP (**Supplementary Figures 30-33 and 39**). As shown in **Supplementary Figure 89** and **Supplementary Table 16**, after introducing the Br atom on the initiators, the phosphorescence proportion in PL spectra showed a dramatic increase up to 91% and 96% for PABr and PBr films, respectively. Benefitting from the heavy atom effect of the Br atom for enhanced SOC and thus promoted intersystem crossing (ISC) process, the RTP quantum yields of PABr and PBr films increased to 35.90% and 13.85%, maintaining a good stretchability of 653.8% and 633.1%, respectively. These data indicated that our proposed multiphase design principle can also prepare stretchable copolymers with high RTP quantum yield, demonstrating the generality of our strategy. The related supplementary data and explanation were added in the revised Supporting Information.

Supplementary Figure 30. Chemical structure of BMP initiator and the ¹H NMR spectrum in CDCl₃.

Supplementary Figure 31. ^{13}C NMR spectrum of BMP initiator in CDCl_3 .

Supplementary Figure 32. Chemical structure of BEP initiator and the ^1H NMR spectrum in CDCl_3 .

Supplementary Figure 33. ^{13}C NMR spectrum of BEP initiator in CDCl_3 .

Supplementary Figure 39. Chemical structures and GPC traces of block copolymers PABr and PPBr, respectively.

Supplementary Figure 89. (a) Normalized steady-state PL (dotted lines) and phosphorescence (solid lines) spectra for PABr and PPBr under ambient conditions. (b) Lifetime decay curves of emission bands at 490 and 585 nm for PABr and PPBr, respectively. (c) Stress-strain curves of PABr and PPBr at room temperature.

Supplementary Table 16. Photoluminescence efficiencies of PABr and PPBr films under ambient conditions.

Polymers	$\lambda_{\text{Fluor.}}$ (nm)	$\Phi_{\text{Fluor.}}$ (%)	$\lambda_{\text{Phos.}}$ (nm)	$\Phi_{\text{Phos.}}$ (%)
PABr	328	1.50	490	35.90
PPBr	413	1.35	585	13.85

Comment #9. The glass-transition values and its reduction upon increasing the chain length is interesting. Authors can attribute the reason for this observation.

Response: We sincerely appreciate the suggestion of the reviewer. Upon increasing the alkyl side chain length, the intermolecular distance and free volume of polymer chains increased, and the molecular interaction weakened, thus the molecular motility increased, which resulted in decreased glass transition temperature and much-improved deformability of corresponding polymer films (*J. Am. Chem. Soc.* **2022**, *144*, 4699; *Macromolecules* **2019**, *52*, 4396; *Nat. Commun.* **2020**, *11*, 893). The related analyses were added in the revised Supporting Information.

Comment #10. How to reduce the glass transition temperature further, authors can state the few techniques through which the reduced glass transition temperature is achievable? In commercial cases, the soft polymeric systems exerts even lower glass transition temperature. How it is important for achieving the good optical properties?

Response: We sincerely appreciate the suggestion of the reviewer. Glass transition is a phenomenon related to the molecular motion of polymer, and molecular structure is the internal condition that determines molecular motion. Therefore, the structure of a polymer chain would have a significant impact on the glass transition temperature (T_g) value. Within a certain range, a longer flexible alkyl side chain would result in a lower T_g due to the larger free volume of polymer chains. Moreover, reducing the steric hindrance of the side group would decrease the block of molecular chain internal rotation, resulting in a lower T_g . Additionally, reducing the number of asymmetric substituents would lower the inside rotating barrier, and hence reduce T_g . However, if the T_g of the soft block is too low, the RTP performance may be weakened. In our designed block copolymers, the PAA hard phase provides a rigid environment for RTP generation, while the poly (alkyl methacrylate) soft phase can enhance polymer chain dynamics for the high stretchability of copolymers. We have demonstrated that the T_g of hard-soft block polymers was constant with that of the second soft block. In other words, the low T_g of block polymers derives from the local motions of the soft poly (alkyl methacrylate) domain. Therefore, when the T_g of the selected soft block is too low, the polymer chain motions will be intense at room temperature, which will cause a strong non-radiative transition and weaken the phosphorescence of copolymers.

To fully demonstrate this state, we introduced hexyl methacrylate with a longer flexible alkyl side chain and butyl acrylate with lower steric hindrance to prepare the soft block. The copolymer structures, glass transition temperature, and photophysical properties were further investigated (**Supplementary Figures 47-48**). Compared with copolymer PABE, polymers PAHE and PABA displayed further reduced T_g of -5 and -54 °C, respectively, and identical phosphorescence emission centers at 548 nm. However, the phosphorescence lifetimes dramatically decreased from 231.36 ms in PABE to 208.59 ms in PAHE and 158.36 ms in PABA films, respectively. Therefore, choosing a soft block with a suitable T_g is crucial for reducing the non-radiative loss from the soft polymeric motions and maintaining the good optical properties of copolymers. Based on these data, the soft block with a T_g close to room temperature is beneficial to achieving RTP copolymers with simultaneously long-lived emission and high stretchability. The related experimental results have been added to the revised Supporting Information.

Supplementary Figure 47. (a) Chemical structures and (b) DSC curves of block copolymers PAHE and PABA.

Supplementary Figure 48. (a) Photoluminescence (black lines) and phosphorescence (red lines) spectra and (b) Lifetime profiles of emission bands at 548 nm for copolymers PAHE and PABA under ambient conditions.

Comment #11. I suggest authors to re-evaluate the stress-strain characteristics.

Response: We are sincerely grateful for the reviewer's suggestion. As suggested by the reviewer, we have carried out five times of parallel mechanical measurements for each sample. There existed some experimental errors for different stress-strain measurements. The error bars have been added in **Figure 2e** in the revised Manuscript, which represents the standard deviation calculated by three times data sets (the highest and lowest data were discarded).

Supporting Figure I. Toughness and Young's modulus variation of PAMA to PABE copolymer films. Error bars represent the standard deviation calculated by three times data sets.

Comment #12. Supplementary Figure 44 graph contrasts the merits of the as-obtained materials. The graphical representation failed to reveal the references in the graphical picture. Add reference in to graph.

Response: We sincerely appreciate the suggestion of this reviewer. As suggested by the reviewer, we have added references to the graph in the revised Supporting Information.

Supporting Figure 52. Performance comparison between this work and reported works⁴⁻¹¹.

Comment #13. Elasticity of the as-obtained phosphorescent composite should be mentioned. Recovery aspects and residual strain characteristics can be studied to differentiate the various systems.

Response: We sincerely appreciate the comments of the reviewer. Taking copolymer PABE-d as an example, we studied the recovery and residual strain properties by cyclic stress-strain tests (**Supporting Figure 53**). The curves showed a pronounced hysteresis, even for a lower applied strain (10%). After the mechanical

stress was removed, only partial mechanical properties of the film could be restored. With increasing the mechanical strain, the residual strain after unloading was larger. The related data and analysis were added in the revised Supporting Information.

Supplementary Figure 53. (a) Stress-strain curves of PABE-d film in successive stretching to different strains under cyclic loading (loading rate: 10 mm min⁻¹). (b) Residual strains after unloading in different cycles.

Comment #14. Upon stretching, lifetime values tend to degrade, why such trends are observed. Authors can add some light to it.

Response: We sincerely appreciate the suggestion of this reviewer. After being stretched, the polymer film showed a considerable residual strain, indicating the sample was not able to recover to its initial state immediately. The rigid molecular structure of the polymer film is possibly affected by mechanical stretching, causing an increase in non-radiative transition. As a result, the phosphorescence lifetime decreased (*Adv. Optical Mater.* **2022**, *10*, 2102449). However, this decrease is only slight, and the polymer film is still able to maintain ultralong RTP after stretching. This suggests good optical stability of the copolymer. The explanation for this phenomenon has been added in the revised Supporting Information.

Comment #15. PABE-d materials exerts underwater phosphorescence. Why this is important, and how the lifetime values are recovered upon annealing at 70°C under vacuum. Why specifically vacuum conditions are imposed, what would be the effects of various humidity or temperatures?

Response: We thank the comments of the reviewer. Generally, for many RTP materials, phosphorescence quenching usually occurs in the presence of water, mainly because of the following reasons: (1) Water molecules have strong polarity and can heavily destroy the hydrogen bonding between chromophores and polymer matrix, resulting in enhanced non-radiative transition. (2) The presence of dissolved oxygen. Triplet excited states of organic molecules are sensitive to surrounding triplet O₂ molecules, which can facilitate triplet-triplet quenching processes. (3) The occurrence of solvent-assisted relaxation or solvent coordination can also result in

considerable phosphorescence quenching especially for metal-based phosphorescent complexes (*Adv. Optical Mater.* 2016, 4, 1397; *Chem. Commun.* **2015**, 51, 10988; *Inorg. Chem.* **2006**, 45, 5721).

In our research, we designed a series of block copolymers to form microphase separation through self-assembly. This strategy not only achieved stretchable RTP materials but also improved their humidity resistance. To demonstrate the advantageous properties of humidity resistance for the resulting copolymers, we compared the phosphorescence properties of PABE-d and HPAA films after soaking them in water for different durations. When soaked in water, the good water solubility of HPAA resulted in the destruction of the hydrogen bonds between polymer chains and chromophores, which led to a dramatic decrease in the RTP intensity of HPAA film (**Supporting Figure 11**). After 25 minutes, the polymer showed nearly no visible RTP phenomenon. However, for block copolymer PABE-d, even though the lifetime and intensity showed a slight decrease, obvious phosphorescence could still be observed even after being immersed in water for 120 minutes (**Supplementary Figure 58**). These results suggest that microphase separation plays a crucial role in preventing water molecules from destroying the hydrogen bonds. This effect is due to the hydrophobic PBMA segment, which can resist and delay the entry of water molecules into the PAA segment and thus protect phosphors, improving the water resistance of these copolymers.

In addition, upon heating at 70 °C under vacuum, the immersed water molecules were removed from the polymer matrix, and the rigid hydrogen-bond networks were restored, thus the ultralong RTP lifetime of PABE-d can be recovered to the initial state. Vacuum conditions can expedite the removal of water molecules from a soaked polymer matrix for restoring the ultralong lifetime of copolymer. However, placing the soaked polymer film in environments with varying humidity or lower temperatures would slow down the removal of water, extending the time required for restoring ultralong phosphorescence lifetime of copolymer.

Supporting Figure 11. (a) Phosphorescence spectra of HPAA film after soaking in water with different times ranging from 0 to 25 min. (b) Phosphorescence intensity of HPAA at 548 nm after soaking in water at varying times. (c) Phosphorescence photographs of HPAA film soaked in water at different times.

Supplementary Figure 58. (a) Phosphorescence spectra and (b) lifetime profiles of the emission band at 548 nm of PABE-d film after being soaked in water with different times ranging from 0 to 120 min. (c) Lifetime profile of the soaked film after heating under vacuum at 70 °C for 2 h.

Comment #16. How the hydrogen bonds contributes to the phosphorescent characteristics? How do authors gauge the extent of hydrogen bond formation within the matrix? Authors can propose the mechanistic scheme for the better understanding.

Response: We are sincerely grateful for the reviewer's suggestion. A mechanistic scheme of the effect of hydrogen bonds on promoting ultralong RTP was presented in **Supplementary Figure 73**: Due to the rich carboxyl groups in the PAA block, there formed multiple inter/intramolecular hydrogen bonds between polymer chains

and chromophores and provided a rigid microenvironment for chromophores. Such a tight and rigid microenvironment helps to confine molecular motions of chromophores and decrease the quenching from surrounding oxygen and moisture, thus stabilizing the triplet state excitons and suppressing non-radiative transition of chromophores for generating ultralong RTP. The related mechanistic scheme was added in the revised Manuscript.

Supplementary Figure 73. Mechanism illustration for the effect of hydrogen bonds on promoting ultralong RTP by multiphase engineering.

The formation of hydrogen bonds can be characterized by FT-IR spectroscopy. Formation of hydrogen bonds would change the stretching of hydrogen donors (N-H, O-H) or electron donors (C=O), and their related infrared absorption peak may shift (*Chem. Mater.* **2019**, *31*, 1430; *Nat. Commun.* **2019**, *10*, 1315; *Nat. Commun.* **2022**, *13*, 4868). We carefully compared the FT-IR spectroscopy of copolymers before and after hydrolysis. As shown in **Supplementary Figure 35**, compared to the FTIR spectra of unhydrolyzed copolymers, the peak corresponding to carbonyl stretching resonance at around 1723 cm^{-1} all showed a red shift to 1703 cm^{-1} in hydrolyzed block copolymers, indicating the formation of $\text{C}=\text{O}\cdots\text{H}-\text{O}$ hydrogen bonds between PAA chains and chromophores. Although we can qualitatively demonstrate the formation of hydrogen bonds within the polymer matrix, we cannot quantitatively analyze the extent of hydrogen bonds at the present stage because of the complicated and unpredictable molecular arrangement between polymer chains and chromophores. We hope the above candid demonstration and explanation can be accepted by the reviewer.

Supplementary Figure 35. (a) FTIR spectra of unhydrolyzed copolymers PBAE, PBME, PBPE, and PBBE (black lines), and hydrolyzed block copolymers PAMA, PAME, PAPE, and PABE (red lines). The characteristic peaks of hydroxy group at around 3100-3600 cm^{-1} indicate that the *tert*-butyl ester groups were converted into carboxyl moieties after hydrolyzation. (b) The shift of carbonyl stretching resonance for copolymers before (black lines) and after (red lines) hydrolyzation.

Comment #17. Absence of clear schematic and mechanism seriously degrades the quality of the presented work. Authors should consider revising the entire article to address the grammar and typographical errors.

Response: We appreciate the critical review from the reviewer for improving the quality of our paper. To give a clear mechanism illustration for better understanding, we have revised **Figure 4g** in the revised Manuscript and added **Supplementary Figure 73** as well as the detailed mechanism explanations in the revised Supporting Information. As suggested by the reviewer, we carefully checked the entire article to correct the grammar and typographical errors. The related modification was highlighted in the revised Manuscript.

Comment #18. I suggest authors to cite some of the recent related literatures. Authors can cite recent references, which can deliver and connect the audience with the current study. Authors can refer the following recently published articles and strengthen introduction and the reference list,

Nature Materials (2023). <https://doi.org/10.1038/s41563-023-01703-0>

Advanced Science, 2021, 8 (21), 2102275. <https://doi.org/10.1002/adv.202102275>

Journal of the American Chemical Society, 2021, 143, 34, 13675–13685.

<https://doi.org/10.1021/jacs.1c05213>

Advanced Materials, 2023, 35, 2303267. <https://doi.org/10.1002/adma.202303267>

ACS Nano 2022, 16, 10, 16201–16210. <https://doi.org/10.1021/acsnano.2c05056>

Chemical Engineering Journal, 418, 2021, 129421.

<https://doi.org/10.1016/j.cej.2021.129421>

ACS Applied Materials & Interfaces, 2023, 15, 28, 33784–33796.

<https://doi.org/10.1021/acsami.3c03794>

ACS Applied Materials & Interfaces, 2020, 12, 12, 14408–14415.

<https://doi.org/10.1021/acsami.9b23291>

Response: We are sincerely grateful for the reviewer's suggestion. As suggested by the reviewer, we have added the relevant references to the revised Manuscript as Ref.38, Ref.43, Ref.23, Ref.4, Ref.37, Ref.42, Ref.9, and Ref.8, respectively. The introduction part was carefully revised as suggested by the reviewer. All the changes were highlighted in the revised Manuscript.

All the best for the revision. After receiving the convincing point-to-point responses, I will recommend it for the publication.

Response: We appreciate the reviewer for the careful review to improve the quality of our paper. We have carefully addressed the concerns raised by the reviewer through a series of additional experiments and detailed explanations. We hope these will make it more acceptable for publication.

REVIEWER COMMENTS

Reviewer #1 (Remarks to the Author):

All the concerns have been well responded and the manuscript has been improved accordingly, the newly added characterizations provide solid support to this work. In my opinion, it can be published.

Reviewer #2 (Remarks to the Author):

Stretchable phosphorescence materials potentially enable applications in diverse advanced fields in wearable electronics. This work presented stretchable phosphorescent materials by combining stiffness and softness simultaneously in well-designed block copolymers. After carefully reading the responses from the authors, this reviewer believed this work should be revised again. In generally, achieving room-temperature phosphorescence simultaneously featuring long-lived emission and high stretchability in some materials would NOT be dilemma since phosphorescence would NOT be influenced ONLY by the 'rigidity'. Therefore, the authors should revise some statements in Introduction and Discussion section to avoid misleading the readers. 1) About tensile rate (5 mm/min @R.T. at now stage) mentioned by the referees, this rate was too slow for further potential applications. The authors should test more fast tensile rates for their systems. 2) About selected NA as chromophore unit in this work, the authors have many statements in their response letter, however, such response statements should be added in the Introduction, and closed-related reference (e.g. JACS, 2018, 140, 1916) should be cited in the revision. 3) About novelty of this work, Referee #2 had listed some previous related publications. But the response from the authors did NOT still demonstrate the novelty and many statements about the differences could not convince. Therefore, the authors should delete some words like 'first', 'highly'. Even for the RTP efficiency as well as stretchability (mentioned by Ref.#1), this work just showed moderate related-performance. 4) About ATRP method used in this work, the end bromine atom at the polymer chain would have still 'heavy atom' affect for phosphorescence? Please discuss this key issue or provide more data to confirm their conclusion.

Reviewer #3 (Remarks to the Author):

Journal: **Nature Communications**

Title: **Stretchable phosphorescent polymers by multiphase engineering**

Authors have successfully demonstrated the fabrication of stretchable block copolymer based phosphorescent materials. Importantly, the as-proposed synthetic methodology provides convenient

optimization of the as-proposed block system with tunable properties. The research carried out seems to have significant impact and the proposed methodology is convincing.

The reply to the comments and revised manuscript is good. I suggest that it can be accepted.

Point-by-Point Response to Reviewers

Reviewer #1:

All the concerns have been well responded and the manuscript has been improved accordingly, the newly added characterizations provide solid support to this work. In my opinion, it can be published.

Response: We sincerely appreciate the reviewer for the positive comments on our manuscript.

Reviewer #2:

Stretchable phosphorescence materials potentially enable applications in diverse advanced fields in wearable electronics. This work presented stretchable phosphorescent materials by combining stiffness and softness simultaneously in well-designed block copolymers. After carefully reading the responses from the authors, this reviewer believed this work should be revised again. In generally, achieving room-temperature phosphorescence simultaneously featuring long-lived emission and high stretchability in some materials would NOT be dilemma since phosphorescence would NOT be influenced ONLY by the 'rigidity'. Therefore, the authors should revise some statements in Introduction and Discussion section to avoid misleading the readers.

Response: We appreciate the reviewer for the careful review to improve the quality of our paper. As suggested by the reviewer, we have revised related statements in the Introduction and Discussion sections. All the changes were highlighted in the revised Manuscript.

Comment #1. About tensile rate (5 mm/min @R.T. at now stage) mentioned by the referees, this rate was too slow for further potential applications. The authors should test more fast tensile rates for their systems.

Response: We are grateful for the suggestion of the reviewer. As suggested, we have taken copolymer PABE-d as a research example and conducted stress-strain curves with faster tensile rates ranging from 10 to 80 mm/min. In addition, to analyze the error, we performed five parallel mechanical measurements for each tensile rate and discarded the highest and lowest data. As shown in **Supplementary Figure 53**, as the tensile rate increased from 10 to 80 mm/min, the strength at break increased, and the strain at break decreased. When the tensile rate was below 50 mm/min, the resulting copolymer could maintain the elongation of beyond 600%. However, when the tensile rate was larger than 50 mm/min, the elongation dropped significantly to 423.6%. We have added the supporting data and relevant illustrations to the revised Manuscript and Supporting Information.

Supplementary Figure 53. (a) Stress-strain curves and (b) Toughness variation of PABE-d film at different tensile rates. Error bars represent the standard deviation calculated by three times data sets.

Supplementary Table 10. Summary of mechanical properties of PABE-d film at different tensile rates.

Tensile rate (mm min ⁻¹)	Young's modulus (MPa)	Strain-at-break (%)	Strength-at-break (MPa)	Toughness (MJ m ⁻³)
10	13.92	665.1	3.87	16.17
20	17.27	650.9	3.92	16.47
30	22.06	629.3	4.18	17.06
40	25.11	620.5	4.21	16.95
50	28.68	600.2	4.49	15.97
80	33.64	423.6	4.85	12.92

Comment #2. About selected NA as chromophore unit in this work, the authors have many statements in their response letter, however, such response statements should be added in the Introduction, and closed-related reference (e.g. JACS, 2018, 140, 1916) should be cited in the revision.

Response: We thank the comments of the reviewer. The reason for selecting naphthalimide derivative as chromophore has been added in the Introduction in the revised Manuscript: A naphthalimide derivative is selected as ATRP initiator and phosphorescence chromophore because the multiple carbonyl groups and nitrogen heteroatom could effectively facilitate the population of triplet states through an efficient ISC process. Moreover, the rigid planar molecular skeletons can restrict molecular motions for suppressed non-radiative decay, contributing to RTP generation. As suggested by the reviewer, the related reference (*J. Am. Chem. Soc.* 2018, 140, 1916-1923) was also cited in the revised Manuscript.

Comment #3. About novelty of this work, Referee #2 had listed some previous related publications. But the response from the authors did NOT still demonstrate the novelty and many statements about the differences could not convince. Therefore, the authors should delete some words like 'first', 'highly'. Even for the

RTP efficiency as well as stretchability (mentioned by Ref.#1), this work just showed moderate related-performance.

Response: We thank the suggestion of the reviewer. As suggested, we have deleted the related illustration. Additionally, all the changes were highlighted in the revised Manuscript and Supporting Information.

Comment #4. About ATRP method used in this work, the end bromine atom at the polymer chain would have still 'heavy atom' affect for phosphorescence? Please discuss this key issue or provide more data to confirm their conclusion.

Response: We truly understand this reviewer's concern. As suggested by the reviewer, to verify whether the terminal bromine atom at the polymer chain have any role in enhancing phosphorescence through the "heavy atom" effect, we further prepared another control polymer **NPAA (without bromine atoms)** through conventional radical polymerization of acrylic acid (AA) and naphthalimide (NA) using 2-azoisobutyronitrile (AIBN) as an initiator (**Supporting Figure 1**). Meanwhile, the photophysical properties of polymer NPAA were also investigated by a set of further experiments. As shown in **Supplementary Figures 43-44**, polymer NPAA displayed similar photophysical properties to polymer HPAA prepared by ATRP, including PL and RTP emission wavelengths, lifetimes, and quantum efficiencies, respectively. Based on these findings, we suggested that the terminal bromine atom present in the polymer chain has an insignificant influence on the RTP performance of the copolymer. Furthermore, the content of terminal bromine atoms on each block copolymer chain is very low. According to the values of n and m presented in **Supplementary Tables 1-2**, the content of bromine atoms in PABEs copolymer systems was approximately in the range of 1/500 to 1/2000. Such a low content will hardly have a heavy atom effect on the chromophores and affect the phosphorescence performance of resulting copolymers. Therefore, the "heavy atom" effect of the terminal bromine atom can be ignored in these block copolymers. The related supporting data and illustration have been added in the revised Manuscript. We hope our candid clarification will be able to convince the reviewer of the importance of our work.

Supporting Figure 1. Synthetic route of polymer NPAA (the molar feed ratio of NA/AA was 1/200).

Supplementary Figure 44. (a) Normalized steady-state photoluminescence (PL) and phosphorescence (Phos.) spectra of NPAA film under 340 nm excitation. (b) Lifetime decay curve of emission band at 548 nm. (c) Lifetime profile of emission band at 395 nm.

Supplementary Figure 43. (a) Normalized steady-state photoluminescence (PL) and phosphorescence (Phos.) spectra of HPAA film under 340 nm excitation. (b) Lifetime decay curve of emission band at 548 nm. (c) Lifetime profile of emission band at 394 nm.

Supplementary Table 6. Photophysical parameters of control polymers NPAA and HPAA under ambient conditions.

Polymer	$\lambda_{\text{Fluor.}}$ (nm)	$\tau_{\text{Fluor.}}$ (ns)	$\Phi_{\text{Fluor.}}$ (%)	$\lambda_{\text{Phos.}}$ (nm)	$\tau_{\text{Phos.}}$ (ms)	$\Phi_{\text{Phos.}}$ (%)
NPAA	395	2.10	11.43	548	270.10	3.17
HPAA	394	2.21	15.12	548	277.17	2.88

Supplementary Table I. GPC characterizations of copolymers PBAE to PBBE. n and m represent the degree of polymerization for each block, which was calculated by M_n .

Sample	PBAE	PBME	PBPE	PBBE
M_w	71110	75520	83060	85230
M_n	56510	59060	61320	68800
PDI	1.25	1.28	1.29	1.24
n	210	210	210	210
m	293	280	267	293
n/m	1:1.4	1:1.3	1:1.3	1:1.4

Supplementary Table 2. GPC characterizations of copolymers PBBEs. n and m represent the degree of polymerization for each block, which was calculated by M_n .

Sample	PBBE-a	PBBE-b	PBBE-c	PBBE-d
M_w	126660	185330	265440	328840
M_n	105120	163290	222620	283800
PDI	1.20	1.13	1.19	1.16
n	210	210	210	210
m	548	958	1375	1805
n/m	1:2.6	1:4.6	1:6.6	1:8.6

Reviewer #3:

Authors have successfully demonstrated the fabrication of stretchable block copolymer based phosphorescent materials. Importantly, the as-proposed synthetic methodology provides convenient optimization of the as-proposed block system with tunable properties. The research carried out seems to have significant impact and the proposed methodology is convincing.

The reply to the comments and revised manuscript is good. I suggest that it can be accepted.

Response: We sincerely appreciate the reviewer for the positive comments on our manuscript.

REVIEWERS' COMMENTS

Reviewer #2 (Remarks to the Author):

This revision could be accepted for publication.